# Neural Contextual Bandits with Upper Confidence Bound-Based Exploration

## Abstract

We study the stochastic contextual bandit problem, where the reward is generated from an unknown bounded function with additive noise. We propose the NeuralUCB algorithm, which leverages the representation power of deep neural networks and uses a neural network-based random feature mapping to construct an upper confidence bound (UCB) of reward for efficient exploration. We prove that, under mild assumptions, NeuralUCB achieves $\widetilde{O}(\sqrt{T})$ regret, where $T$ is the number of rounds. To the best of our knowledge, our algorithm is the first neural network-based contextual bandit algorithm with near-optimal regret guarantee. Preliminary experiment results on synthetic data corroborate our theory, and shed light on potential applications of our algorithm to real-world problems.

## 1 Introduction

The stochastic contextual bandit problem has been extensively studied in machine learning (Bubeck and Cesa-Bianchi, 2012; Lattimore and Szepesvári, 2019): at round $t \in \{1, 2, \ldots, T\}$, an agent is presented with a set of $K$ actions, each of which is associated with a $d$-dimensional feature vector. After choosing an action, the agent will receive a stochastic reward generated from some unknown distribution conditioned on the chosen action's feature vector. The goal of the agent is to maximize the expected cumulative rewards over $T$ rounds. Contextual bandit algorithms have been applied to many real-world applications, such as personalized recommendation, advertising and Web search (e.g., Agarwal et al., 2009; Li et al., 2010).

The most studied model in the literature is linear contextual bandits (Auer, 2002; Abe et al., 2003; Dani et al., 2008; Rusmevichientong and Tsitsiklis, 2010; Chu et al., 2011; Abbasi-Yadkori et al., 2011), which assumes that the expected reward at each round is a linear function of the feature vector. Linear bandit algorithms have achieved great success in both theory and practice, such as news article recommendation (Li et al., 2010). However, the linear-reward assumption often fails to hold exactly in practice, which motivates the study of nonlinear contextual bandits (e.g., Filippi et al., 2010; Srinivas et al., 2010; Bubeck et al., 2011; Valko et al., 2013). However, they still require fairly strong assumptions on the reward function. For instance, Filippi et al. (2010) makes a generalized linear model assumption on the reward, Bubeck et al. (2011) require it to have a Lipschitz continuous property in a proper metric space, and Valko et al. (2013) assume the reward function belongs to some Reproducing Kernel Hilbert Space (RKHS).

In order to overcome the above shortcomings, deep neural networks (DNNs) (Goodfellow et al., 2016) have been introduced to learn the underlying reward function in contextual bandit problem, thanks to their strong representation power. Given the fact that DNNs enable the agent to make use of nonlinear models with less domain knowledge, existing work (Riquelme et al., 2018; Zahavy and Mannor, 2019) focuses on the idea called *neural-linear bandit*. More precisely, they use the first $L - 1$ layers of a DNN as a feature map, which transforms contexts from the raw input space to a low-dimensional space, usually with better representation and less frequent update. Then they learn a linear exploration policy on top of the last hidden layer of the DNN with a more frequent update. These attempts have achieved great empirical success. However, none of these work provides a theoretical guarantee on the regret of the algorithms.

In this paper, we take the first step towards provable efficient contextual bandit algorithms based on deep neural networks. Specifically, we propose a new algorithm, NeuralUCB, which uses a deep neural network to learn the underlying reward function. At the core of the algorithm is an upper

confidence bound constructed by deep neural network-based random feature mappings. Our regret analysis of NeuralUCB is built on recent results on optimization and generalization of deep neural networks (Jacot et al., 2018; Arora et al., 2019; Cao and Gu, 2019a). While the main focus of our paper is mostly theoretical, we also carry out proof-of-concept experiments on synthetic data to validate the effectiveness of our proposed algorithm.

Our contributions are summarized as follows:

- We prove that, under mild assumptions, our algorithm is able to achieve a $\widetilde{O}(\widetilde{d}\sqrt{T})$ regret, where $\widetilde{d}$ is the effective dimension of a neural tangent kernel matrix and $T$ is the number of rounds. Our regret bound recovers the $\widetilde{O}(d\sqrt{T})$ regret for linear contextual bandit as a special case (Abbasi-Yadkori et al., 2011), where $d$ is the dimension of context.

- We propose a neural contextual bandit algorithm using neural network-based exploration. It can be regarded as an extension of existing linear bandit algorithms (Li et al., 2010; Abbasi-Yadkori et al., 2011), from linear reward functions to any bounded reward functions.

- We provide empirical evidence in several proof-of-concept experiments to demonstrate potential applications of our algorithm to real-world problems.

**Notation:** Scalars are denoted by lower case letters, vectors by lower case bold face letters, and matrices by upper case bold face letters. For a positive integer $k$, $[k]$ denotes $\{1, \ldots, k\}$. For a vector $\boldsymbol{\theta} \in \mathbb{R}^d$, we denote its $\ell_2$ norm by $\|\boldsymbol{\theta}\|_2 = \sqrt{\sum_{i=1}^d \theta_i^2}$ and its $j$-th coordinate by $[\boldsymbol{\theta}]_j$. For a matrix $\mathbf{A} \in \mathbb{R}^{d \times d}$, we denote its spectral norm, Frobenius norm, and $(i, j)$-th entry by $\|\mathbf{A}\|_2, \|\mathbf{A}\|_F$, and $[\mathbf{A}]_{i,j}$, respectively. We denote a sequence of vectors by $\{\boldsymbol{\theta}_j\}_{j=1}^t$, and similarly for matrices. For two sequences $\{a_n\}$ and $\{b_n\}$, we use $a_n = O(b_n)$ to denote that there exists some constant $C > 0$ such that $a_n \le Cb_n$, $a_n = \Omega(b_n)$ to denote that there exists some constant $C' > 0$ such that $a_n \ge C'b_n$. In addition, we use $\widetilde{O}(\cdot)$ to hide logarithmic factors. We say a random variable $X$ is $\nu$-sub-Gaussian if $\mathbb{E}\exp(\lambda(X - \mathbb{E}X)) \le \exp(\lambda^2 \nu^2 / 2)$ for any $\lambda > 0$.

## 2 RELATED WORK

### 2.1 CONTEXTUAL BANDITS

There is a line of extensive work on linear bandits (e.g., Auer, 2002; Abe et al., 2003; Dani et al., 2008; Rusmevichientong and Tsitsiklis, 2010; Li et al., 2010; Chu et al., 2011; Abbasi-Yadkori et al., 2011). For the setting with finitely many arms, Abe et al. (2003) formalized the linear bandit setting and analyzed some of the earliest algorithms. Auer (2002) proposed SupLinRel algorithm that achieves $\widetilde{O}\sqrt{dT}$ regret. Chu et al. (2011) obtained the same regret with SupLinUCB that is based on LinUCB (Li et al., 2010); the authors also provided a lower bound of $\Omega(\sqrt{dT})$. For the other setting with infinitely many arms, a few authors (Dani et al., 2008; Rusmevichientong and Tsitsiklis, 2010; Abbasi-Yadkori et al., 2011) proposed algorithms that achieve $\widetilde{O}(d\sqrt{T})$ regret, whereas an $\Omega(d\sqrt{T})$ lower bound is given by Dani et al. (2008).

While most algorithms above are based on the idea of upper confidence bounding, it is also possible to use proper randomization to achieve strong regret guarantees, such as Thompson sampling and reward perturbation (Thompson, 1933; Chapelle and Li, 2011; Agrawal and Goyal, 2013; Russo and Van Roy, 2014; 2016; Kveton et al., 2019).

To deal with nonlinearity, generalized linear bandit has been considered, which assumes that the reward function can be written as a composition of a linear function and a link function. In particular, Filippi et al. (2010) proposed a GLM-UCB algorithm, which attains $\widetilde{O}(d\sqrt{T})$ regret. Li et al. (2017) proposed SupCB-GLM for generalized contextual bandit problems and showed a $\widetilde{O}(\sqrt{dT})$ regret that matches the lower bound. Jun et al. (2017) studied how to scale up algorithms for GLM bandits.

A few authors have also explored more general nonlinear bandits without making strong modeling assumptions. One line of work is variants of expert learning algorithms (Auer et al., 2002), which typically have a time complexity linear in the number of experts (which in many cases can be exponential in the number of parameters). Another approach is to reduce a bandit problem to supervised

learning, such as the epoch-greedy algorithm (Langford and Zhang, 2008) that has an $O(T^{2/3})$ regret. Later, Agarwal et al. (2014) develop an algorithm that yields a near-optimal regret bound, but relies on an optimization oracle that can be expensive. A third approach uses nonparametric modeling, such as Gaussian processes and kernels (Srinivas et al., 2010; Krause and Ong, 2011; Valko et al., 2013). More specifically, Srinivas et al. (2010) assumed that the reward function is generated from a Gaussian process with known mean and covariance functions. They proposed a GP-UCB algorithm which achieves $\widetilde{O}(\sqrt{T\gamma_T})$ regret, where $\gamma_T$ is the maximum information gain. Krause and Ong (2011) assumed the reward function is defined over the context-arm joint space and proposed a Contextual GP-UCB. Valko et al. (2013) assumed that the reward function lies in a RKHS with bounded RKHS norm. They proposed a SupKernelUCB algorithm and showed an $\widetilde{O}(\sqrt{\widetilde{d}T})$ regret, where $\widetilde{d}$ is effective dimension of the kernel that can be seen as a generalized notion of the dimension of contexts. There is also work focusing on bandit problems in general metric space with Lipschitz continuous property on the context (Kleinberg et al., 2008; Bubeck et al., 2011).

## 2.2 NEURAL NETWORKS

Different lines of research have been done to provide theoretical understandings of DNNs from different aspects. For example, to understand how the expressive power of DNNs are related to their architecture, Telgarsky (2015; 2016); Liang and Srikant (2016); Yarotsky (2017; 2018); Hanin (2017) showed that deep neural networks can express more function classes than shallow networks. Lu et al. (2017); Hanin and Sellke (2017) suggested that the width of neural networks is crucial to improve the expressive power of neural networks.

For the optimization of DNNs, a series of work have been proposed to show that (stochastic) gradient descent can find the global minima of training loss (Li and Liang, 2018; Du et al., 2019b; Allen-Zhu et al., 2019; Du et al., 2019a; Zou et al., 2019; Zou and Gu, 2019). For the generalization of DNNs, a series of work (Daniely, 2017; Cao and Gu, 2019a;b; Arora et al., 2019) shows that by using (stochastic) gradient descent, the parameters of a DNN are located in a particular regime and the generalization bound of DNNs can be characterized by the best function in the corresponding neural tangent kernel space (Jacot et al., 2018).

## 3 PROBLEM SETTING

We consider the stochastic $K$-armed contextual bandit problem, where the total number of rounds $T$ is known. At round $t \in [T]$, the agent observes the $t$th context consisting of $K$ feature vectors: $\{\mathbf{x}_{t,a} \in \mathbb{R}^d \mid a \in [K]\}$. The agent selects an action $a_t$ and receive a reward $r_{t,a_t}$. For simplicity, we denote $\{\mathbf{x}^i\}_{i=1}^{TK}$ as the collection of $\{\mathbf{x}_{1,1}, \mathbf{x}_{1,2}, \ldots, \mathbf{x}_{T,K}\}$. Our goal is to maximize the following *pseudo regret* (or *regret* for short):

$$R_T = \mathbb{E}\left[\sum_{t=1}^{T}(r_{t,a_t^*} - r_{t,a_t})\right], \tag{3.1}$$

where $a_t^*$ is the optimal action at round $t$ that maximizes the expected reward, i.e., $a_t^* = \operatorname{argmax}_{a \in [K]} \mathbb{E}[r_{t,a}]$.

This work makes the following assumption on reward generation: for any round $t$,

$$r_{t,a_t} = h(\mathbf{x}_{t,a_t}) + \xi_t, \tag{3.2}$$

where $h$ is some unknown function satisfying $0 \le h(\mathbf{x}) \le 1$ for any $\mathbf{x}$, and $\xi_t$ is $\nu$-sub-Gaussian noise conditioned on $\mathbf{x}_{1,a_1}, \ldots, \mathbf{x}_{t-1,a_{t-1}}$. Note that the $\nu$-sub-Gaussian noise assumption for $\xi_t$ is a standard assumption in stochastic bandit literature (e.g., Abbasi-Yadkori et al., 2011); in particular, any bounded noise satisfies such an assumption. Our reward function class contains linear functions, generalized linear functions, Gaussian processes, and kernel functions with bounded RKHS norm over a bounded domain.

In order to learn the reward function $h$ in (3.2), we propose to use a fully connected deep neural networks with depth $L \ge 2$:

$$f(\mathbf{x}; \boldsymbol{\theta}) = \sqrt{m}\mathbf{W}_L\sigma\Big(\mathbf{W}_{L-1}\sigma\big(\cdots\sigma(\mathbf{W}_1\mathbf{x})\big)\Big), \tag{3.3}$$

where $\sigma(x) = \max\{x, 0\}$ is the rectified linear unit (ReLU) activation function, $\mathbf{W}_1 \in \mathbb{R}^{m \times d}$, $\mathbf{W}_i \in \mathbb{R}^{m \times m}$, $2 \leq i \leq L - 1$, $\mathbf{W}_L \in \mathbb{R}^{m \times 1}$, and $\boldsymbol{\theta} = [\text{vec}(\mathbf{W}_1)^\top, \ldots, \text{vec}(\mathbf{W}_L)^\top]^\top \in \mathbb{R}^p$ with $p = m + md + m^2(L - 1)$. Without loss of generality, we assume that the width of each hidden layer is the same (i.e., $m$) for convenience in analysis. We denote the gradient of the neural network function by $\mathbf{g}(\mathbf{x}; \boldsymbol{\theta}) = \nabla_{\boldsymbol{\theta}} f(\mathbf{x}; \boldsymbol{\theta}) \in \mathbb{R}^p$.

## 4 THE NEURALUCB ALGORITHM

We present in Algorithm 1 our algorithm, NeuralUCB. The key idea is to use a deep neural network $f(\mathbf{x}; \boldsymbol{\theta})$ to predict the reward of context $\mathbf{x}$, and upper confidence bound-based exploration (Auer, 2002). In particular, Algorithm 1 first initializes the network by randomly generating each entry of $\boldsymbol{\theta}$ from an appropriate Gaussian distribution. More specifically, for each $1 \leq l \leq L - 1$, we initialize $\mathbf{W}_l$ as $\begin{pmatrix} \mathbf{W} & \mathbf{0} \\ \mathbf{0} & \mathbf{W} \end{pmatrix}$, where each entry of $\mathbf{W}$ is generated independently from $N(0, 4/m)$. For $\mathbf{W}_L$, we initialize $\mathbf{W}_L$ as $(\mathbf{w}^\top, -\mathbf{w}^\top)$, where each entry of $\mathbf{w}$ is generated independently from $N(0, 2/m)$. At round $t$, Algorithm 1 observes the contexts for all actions, $\{\mathbf{x}_{t,a}\}_{a=1}^K$. First, it computes the upper confidence bound, $U_{t,a}$, based on context $\mathbf{x}_{t,a}$, the current neural network parameter $\theta_{t-1}$, and a scaling factor $\gamma_{t-1}$. It then chooses action $a_t$ with the largest $U_{t,a}$, and receives the corresponding reward $r_{t,a_t}$. At the end of round $t$, Algorithm 1 updates $\boldsymbol{\theta}_t$ by applying Algorithm 2 to (approximately) minimize $L(\boldsymbol{\theta})$ using gradient descent, and updates $\gamma_t$. We choose gradient descent in Algorithm 2 for the simplicity of analysis, although the training method can be replaced by more efficient algorithms like stochastic gradient descent with a more involved analysis (Allen-Zhu et al., 2019; Zou et al., 2019) .

---

**Algorithm 1** NeuralUCB

---

1: **Input:** Number of rounds $T$, regularization parameter $\lambda$, exploration parameter $\nu$, confidence parameter $\delta$, norm parameter $S$, step size $\eta$, number of gradient descent steps $J$, network width $m$, network depth $L$.
2: **Initialization:** Randomly initialize $\boldsymbol{\theta}_0$ as described in the text
3: Initialize $\mathbf{Z}_0 = \lambda \mathbf{I}$, $\mathbf{b}_0 = \mathbf{0}$
4: **for** $t = 1, \ldots, T$ **do**
5:     Observe $\{\mathbf{x}_{t,a}\}_{a=1}^K$
6:     **for** $a = 1, \ldots, K$ **do**
7:         Compute $U_{t,a} = f(\mathbf{x}_{t,a}; \boldsymbol{\theta}_{t-1}) + \gamma_{t-1}\sqrt{\mathbf{g}(\mathbf{x}_{t,a}; \boldsymbol{\theta}_{t-1})^\top \mathbf{Z}_{t-1}^{-1} \mathbf{g}(\mathbf{x}_{t,a}; \boldsymbol{\theta}_{t-1})/m}$
8:         Let $a_t = \text{argmax}_{a \in [K]} U_{t,a}$
9:     **end for**
10:    Play $a_t$ and observe reward $r_{t,a_t}$
11:    Let $\boldsymbol{\theta}_t = \text{TrainNN}(\lambda, \eta, J, m, \{\mathbf{x}_{i,a_i}\}_{i=1}^t, \{r_{i,a_i}\}_{i=1}^t, \boldsymbol{\theta}_0)$
12:    Compute $\mathbf{Z}_t = \mathbf{Z}_{t-1} + \mathbf{g}(\mathbf{x}_{t,a_t}; \boldsymbol{\theta}_t)\mathbf{g}(\mathbf{x}_{t,a_t}; \boldsymbol{\theta}_t)^\top/m$
13:    Compute

$$\gamma_t = \sqrt{1 + C_1 m^{-1/6}\sqrt{\log m} L^4 t^{7/6} \lambda^{-7/6}}$$
$$\cdot \left( \nu \sqrt{\log \frac{\det \mathbf{Z}_t}{\det \lambda \mathbf{I}} + C_2 m^{-1/6}\sqrt{\log m} L^4 t^{5/3} \lambda^{-1/6} - 2\log \delta} + \sqrt{\lambda} S \right)$$
$$+ C_3 \left[ (1 - \eta m \lambda)^J \sqrt{t/\lambda} + m^{-1/6}\sqrt{\log m} L^{7/2} t^{5/3} \lambda^{-5/3}(1 + \sqrt{t/\lambda}) \right].$$

14: **end for**

---

**Comparison with Existing Algorithms**   Here, we compare NeuralUCB with other neural network based contextual bandit algorithms. Allesiardo et al. (2014) proposed NeuralBandit which consists of $K$ neural networks. It uses a committee of networks to compute the score of each action and chooses the action by $\epsilon$-greedy policy. In contrast, our NeuralUCB uses upper confidence bound based exploration, which is more effective than $\epsilon$-greedy. In addition, our algorithm only uses one neural network instead of $K$ neural networks, thus can be computationally more efficient.

---

**Algorithm 2** TrainNN$(\lambda, \eta, J, m, \{\mathbf{x}_{i,a_i}\}_{i=1}^t, \{r_{i,a_i}\}_{i=1}^t, \boldsymbol{\theta}^{(0)})$

---

1: **Input:** Regularization parameter $\lambda$, step size $\eta$, number of gradient descent steps $U$, network width $m$, contexts $\{\mathbf{x}_{i,a_i}\}_{i=1}^t$, rewards $\{r_{i,a_i}\}_{i=1}^t$, initial parameter $\boldsymbol{\theta}^{(0)}$.
2: Define $L(\boldsymbol{\theta}) = \sum_{i=1}^t (f(\mathbf{x}_{i,a_i}; \boldsymbol{\theta}) - r_{i,a_i})^2/2 + m\lambda\|\boldsymbol{\theta} - \boldsymbol{\theta}^{(0)}\|_2^2/2$.
3: **for** $j = 0, \ldots, J-1$ **do**
4: $\quad \boldsymbol{\theta}^{(j+1)} = \boldsymbol{\theta}^{(j)} - \eta\nabla L(\boldsymbol{\theta}^{(j)})$
5: **end for**
6: **return** $\boldsymbol{\theta}^{(J)}$.

---

Lipton et al. (2018) used Thompson sampling on deep neural networks (through variational inference) in reinforcement learning. A variant is proposed by Azizzadenesheli et al. (2018) that works well on a set of Atari benchmarks. Riquelme et al. (2018) proposed NeuralLinear, which uses the first $L-1$ layers of a $L$-layer DNN to learn a representation, then applies Thompson sampling on the last layer to choose action. Zahavy and Mannor (2019) proposed a NeuralLinear with limited memory (NeuralLinearLM), which also uses the first $L-1$ layers of a $L$-layer DNN to learn a representation and applies Thompson sampling on the last layer. Instead of computing the exact mean and variance in Thompson sampling, NeuralLinearLM only computes their approximation. Unlike NeuralLinear and NeuralLinearLM, NeuralUCB uses the entire DNN to learn the representation and constructs the upper confidence bound based on the random feature mapping defined by the neural network gradient.

**Variant of NeuralUCB** We provide a variant of NeuralUCB called NeuralUCB$_0$ in Appendix E. NeuralUCB$_0$ can be regarded as a simplified version of NeuralUCB where the neural network parameter vector $\boldsymbol{\theta}_t$ is not updated at each round. In this sense, NeuralUCB$_0$ can be seen as KernelUCB (Valko et al., 2013) specialized to the Neural Tangent Kernel (Jacot et al., 2018), or LinUCB (Li et al., 2010) with Neural Tangent Random Features (Cao and Gu, 2019a). We will include both NeuralUCB and NeuralUCB$_0$ in the experiments.

**Efficient Implementation** Algorithm 1 can be implemented efficiently using iterative first-order algorithms, as in linear bandit literature (Li et al., 2010; Valko et al., 2013; Agarwal et al., 2014). More details are given here for the sake of completeness. At round $t$, we define $\mathbf{d}_{t,a} = \mathbf{Z}_{t-1}^{-1}\mathbf{g}(\mathbf{x}_{t,a}; \boldsymbol{\theta}_t)$. Based on $\mathbf{d}_{t,a}$, we update $\det(\mathbf{Z}_t)$ by matrix determinant lemma (Golub and Van Loan, 1996):

$$\det(\mathbf{Z}_t) = \det\left[\mathbf{Z}_{t-1} + \mathbf{g}(\mathbf{x}_{t,a_t}; \boldsymbol{\theta}_t)\mathbf{g}(\mathbf{x}_{t,a_t}; \boldsymbol{\theta}_t)^\top/m\right]$$
$$= \left[1 + \mathbf{g}(\mathbf{x}_{t,a_t}; \boldsymbol{\theta}_t)^\top\mathbf{Z}_{t-1}^{-1}\mathbf{g}(\mathbf{x}_{t,a_t}; \boldsymbol{\theta}_t)/m\right]\det(\mathbf{Z}_{t-1}). \tag{4.1}$$

Note that (4.1) only requires to compute vector inner product between $\mathbf{g}(\mathbf{x}_{t,a_t}; \boldsymbol{\theta}_t)$ and $\mathbf{d}_{t,a}$, which only requires $O(p)$ time. Now we show how to compute $\mathbf{d}_{t,a}$ efficiently. By the definition of $\mathbf{d}_{t,a}$, we have

$$\mathbf{g}(\mathbf{x}_{t,a}; \boldsymbol{\theta}_t) = \mathbf{Z}_{t-1}\mathbf{d}_{t,a} = \left(\lambda\mathbf{I} + \sum_{i=0}^{t-1} \mathbf{g}(\mathbf{x}_{i,a_i}; \boldsymbol{\theta}_i)\mathbf{g}(\mathbf{x}_{i,a_i}; \boldsymbol{\theta}_i)^\top/m\right)\mathbf{d}_{t,a}.$$

Thus, $\mathbf{d}_{t,a}$ is the global minimizer of the following convex optimization problem:

$$\min_{\mathbf{d}\in\mathbb{R}^p} \left\|\left(\lambda\mathbf{I} + \sum_{i=0}^{t-1} \mathbf{g}(\mathbf{x}_{i,a_i}; \boldsymbol{\theta}_i)\mathbf{g}(\mathbf{x}_{i,a_i}; \boldsymbol{\theta}_i)^\top/m\right)\mathbf{d} - \mathbf{g}(\mathbf{x}_{t,a}; \boldsymbol{\theta}_t)\right\|_2^2,$$

and we can use (stochastic) gradient descent to find $\mathbf{d}_{t,a}$ efficiently, whose update formula can be written as follows:

$$\mathbf{d}^{(i+1)} = \mathbf{d}^{(i)} - \eta\left(\lambda\mathbf{I} + \sum_{i=0}^{t-1} \mathbf{g}(\mathbf{x}_{i,a_i}; \boldsymbol{\theta}_i)\mathbf{g}(\mathbf{x}_{i,a_i}; \boldsymbol{\theta}_i)^\top/m\right)$$
$$\left[\left(\lambda\mathbf{I} + \sum_{i=0}^{t-1} \mathbf{g}(\mathbf{x}_{i,a_i}; \boldsymbol{\theta}_i)\mathbf{g}(\mathbf{x}_{i,a_i}; \boldsymbol{\theta}_i)^\top/m\right)\mathbf{d}^{(i)} - \mathbf{g}(\mathbf{x}_{t,a}; \boldsymbol{\theta}_t)\right],$$

which again only requires to compute vector inner products.

## 5   Regret Analysis

In this section, we present a regret analysis for Algorithm 1. Recall that $\{\mathbf{x}^i\}_{i=1}^{TK}$ is the collection of all $\{\mathbf{x}_{t,a}\}$. Since our regret analysis is built upon the recently proposed neural tangent kernel matrix (Jacot et al., 2018), we start with its formal definition.

**Definition 5.1** (Jacot et al. (2018); Cao and Gu (2019a)). For a set of contexts $\{\mathbf{x}^i\}_{i=1}^{TK}$, define

$$\widetilde{\mathbf{H}}_{i,j}^{(1)} = \mathbf{\Sigma}_{i,j}^{(1)} = \langle \mathbf{x}^i, \mathbf{x}^j \rangle, \qquad \mathbf{A}_{i,j}^{(l)} = \begin{pmatrix} \mathbf{\Sigma}_{i,i}^{(l)} & \mathbf{\Sigma}_{i,j}^{(l)} \\ \mathbf{\Sigma}_{i,j}^{(l)} & \mathbf{\Sigma}_{j,j}^{(l)} \end{pmatrix},$$

$$\mathbf{\Sigma}_{i,j}^{(l+1)} = 2\mathbb{E}_{(u,v)\sim N(\mathbf{0},\mathbf{A}_{i,j}^{(l)})}\sigma(u)\sigma(v),$$

$$\widetilde{\mathbf{H}}_{i,j}^{(l+1)} = 2\widetilde{\mathbf{H}}_{i,j}^{(l)}\mathbb{E}_{(u,v)\sim N(\mathbf{0},\mathbf{A}_{i,j}^{(l)})}\sigma'(u)\sigma'(v) + \mathbf{\Sigma}_{i,j}^{(l+1)}.$$

Then, $\mathbf{H} = (\widetilde{\mathbf{H}}^{(L)} + \mathbf{\Sigma}^{(L)})/2$ is called the *neural tangent kernel (NTK)* matrix on the context set.

As shown in Definition 5.1, the Gram matrix of the NTK on the contexts $\{\mathbf{x}^i\}_{i=1}^{TK}$ for $L$-layer neural networks is defined in a recursive way from the input layer all the way to the output layer of the neural network. For more details about the derivation of Definition 5.1, please refer to Jacot et al. (2018). Based on Definition 5.1, we first lay out the assumption on the contexts $\{\mathbf{x}^i\}_{i=1}^{TK}$.

**Assumption 5.2.** $\mathbf{H} \succeq \lambda_0 \mathbf{I}$. Moreover, for any $1 \le i \le TK$, $\|\mathbf{x}^i\|_2 = 1$ and $[\mathbf{x}^i]_j = [\mathbf{x}^i]_{j+d/2}$.

The first part of the assumption says that the neural tangent kernel matrix is non-singular, a mild assumption commonly made in the related literature (Du et al., 2019a; Arora et al., 2019; Cao and Gu, 2019a). It can be satisfied as long as no two contexts in $\{\mathbf{x}^i\}_{i=1}^{TK}$ are parallel. The second part is also mild: for any context $\mathbf{x}$, $\|\mathbf{x}\|_2 = 1$, we can always construct a new context $\mathbf{x}' = [\mathbf{x}^\top, \mathbf{x}^\top]^\top/\sqrt{2}$ to satisfy Assumption 5.2. It can be verified that if $\boldsymbol{\theta}_0$ is generated by the random initialization scheme in Algorithm 1, then $f(\mathbf{x}^i; \boldsymbol{\theta}_0) = 0$ for any $i \in [TK]$.

Next we define the effective dimension $\widetilde{d}$ of the neural tangent kernel matrix on contexts $\{\mathbf{x}^i\}_{i=1}^{TK}$.

**Definition 5.3.** The effective dimension $\widetilde{d}$ of the neural tangent kernel matrix on contexts $\{\mathbf{x}^i\}_{i=1}^{TK}$ is defined as

$$\widetilde{d} = \frac{\log\det(\mathbf{I} + \mathbf{H}/\lambda)}{\log(1 + TK/\lambda)}. \tag{5.1}$$

**Remark 5.4.** The notion of effective dimension was introduced by Valko et al. (2013) for analyzing kernel contextual bandits, which was defined by the eigenvalues of any kernel matrix restricted on the given contexts. We adapt a similar but different definition of Yang and Wang (2019), which was used for the analysis of kernel-based Q-learning. Suppose the effective dimension of the reproducing kernel Hilbert space induced by the given kernel is $\widehat{d}$ and the feature mapping $\boldsymbol{\psi}$ induced by the given kernel satisfies $\|\boldsymbol{\psi}(\mathbf{x})\|_2 \le 1$ for any $\mathbf{x} \in \mathbb{R}^d$. Then, it is easy to verify that we always have $\widetilde{d} \le \widehat{d}$; see Appendix A.1 for details. Intuitively, $\widetilde{d}$ measures how quickly the eigenvalues of $\mathbf{H}$ decay, and it only depends on $T$ logarithmically in certain specific cases (Valko et al., 2013).

Now we are ready to present the main result, which provides the regret bound $R_T$ of Algorithm 1.

**Theorem 5.5.** Let $\widetilde{d}$ be the effective dimension defined in Definition 5.3. Let $\mathbf{h} = [h(\mathbf{x}^i)]_{i=1}^{TK} \in \mathbb{R}^{TK}$. Suppose there are constants $C_1, C_2 > 0$, such that for any $\delta \in (0, 1)$, if

$$m = \text{poly}(T, L, K, \lambda^{-1}, \lambda_0^{-1}, S^{-1}, \log(1/\delta)), \quad \eta = C_1(mTL + m\lambda)^{-1},$$

$\lambda \ge \max\{1, S^{-2}\}$, and $S \ge \sqrt{2\mathbf{h}^\top \mathbf{H}^{-1}\mathbf{h}}$. Then, with probability at least $1 - \delta$ over the random initialization of $\boldsymbol{\theta}_0$, the regret of Algorithm 1 satisfies

$$R_T \le 3\sqrt{T}\sqrt{\widetilde{d}\log(1 + TK/\lambda) + 2}\left[\nu\sqrt{\widetilde{d}\log(1 + TK/\lambda) + 2 - 2\log\delta}\right.$$

$$\left. + 2\sqrt{\lambda}S + C_2(1 - \lambda/(TL))^J\sqrt{T/\lambda}\right] + 1. \tag{5.2}$$

**Remark 5.6.** It is worth noting that, simply applying results for linear bandit to our algorithm would lead to a linear dependence of $p$ or $\sqrt{p}$ in the regret. Such a bound is vacuous since in our setting $p$ would be very large compared with the number of rounds $T$ and the input context dimension $d$. In contrast, our regret bound only depends on $\widetilde{d}$, which is much smaller than $p$.

**Remark 5.7.** Our regret bound (5.2) has a term $(1 - \lambda/(TL))^J \sqrt{T/\lambda}$, which characterizes the optimization error of Algorithm 2. More specifically, it suffices to set $J = \log(\lambda S/\sqrt{T})/(\lambda/(TL)) = \widetilde{O}(TL/\lambda)$ (independent of $m$) such that $(1 - \lambda/(TL))^J \sqrt{T/\lambda} \leq \sqrt{\lambda}S$, and therefore the optimization error is dominated by $\sqrt{\lambda}S$.

**Remark 5.8.** Treating $\nu$ and $\lambda$ as constants, and taking $S = \sqrt{2\mathbf{h}^\top \mathbf{H}^{-1}\mathbf{h}}$ and $J = \log(\lambda S/\sqrt{T})/(\lambda/(TL))$, the regret bound (5.2) becomes $R_T = \widetilde{O}\Big(\sqrt{\widetilde{d}T}\sqrt{\max\{\widetilde{d}, \mathbf{h}^\top \mathbf{H}^{-1}\mathbf{h}\}}\Big)$. Specifically, suppose $h$ belongs to a RKHS $\mathcal{H}$ induced by NTK and it can be represented by $h(\mathbf{x}) = \langle \psi(\mathbf{x}), \boldsymbol{\beta}^* \rangle$, where $\psi(\mathbf{x})$ is the feature mapping induced by NTK, for some $\boldsymbol{\beta}^*$ in $\mathcal{H}$ (Valko et al., 2013). Then we can show that $\|h\|_{\mathcal{H}} = \|\boldsymbol{\beta}^*\|_2 \geq \sqrt{\mathbf{h}^\top \mathbf{H}^{-1}\mathbf{h}}$; see Appendix A.2 for more details. Thus our regret bound can be further written as $R_T = \widetilde{O}\Big(\sqrt{\widetilde{d}T}\sqrt{\max\{\widetilde{d}, \|h\|_{\mathcal{H}}\}}\Big)$.

The high-probability result in Theorem 5.5 can be used to obtain a bound on the expected regret.

**Corollary 5.9.** Under the same conditions in Theorem 5.5, we have

$$\mathbb{E}[R_T] \leq 3\sqrt{T}\sqrt{\widetilde{d}\log(1 + TK/\lambda) + 2}\left[\nu\sqrt{\widetilde{d}\log(1 + TK/\lambda) + 2 + 2\log T}\right.$$
$$\left. + 2\sqrt{\lambda}S + C_1(1 - \lambda/(TL))^J\sqrt{T/\lambda}\right] + 2.$$

## 6  PROOF OF MAIN RESULTS

This section provides the proof of Theorem 5.5. We first point out several technical challenges in this proof:

- We do not make parametric assumptions on the reward function as some previous work (Filippi et al., 2010; Chu et al., 2011; Abbasi-Yadkori et al., 2011). Furthermore, unlike the fixed feature mapping used in Valko et al. (2013), NeuralUCB uses neural network $f(\mathbf{x}; \boldsymbol{\theta}_t)$ and its gradient $\mathbf{g}(\mathbf{x}; \boldsymbol{\theta}_t)$ as a dynamic feature mapping depending on $\boldsymbol{\theta}_t$. These differences make the regret analysis of NeuralUCB more difficult.

- In practice, the neural network is often overparametrized, which implies $m$ (and thus $p$) is very large. Thus, we need to make sure the regret bound is independent of $m$.

The two challenges above are addressed by the following technical lemmas. Their proofs are gathered in the appendix.

**Lemma 6.1.** There exists some constant $\bar{C} > 0$ such that for any $\delta \in (0, 1)$, if $m \geq \bar{C}T^4K^4L^6\log(T^2K^2L/\delta)/\lambda_0^4$, then with probability at least $1 - \delta$ over the random initialization of $\boldsymbol{\theta}_0$, there is a $\boldsymbol{\theta}^* \in \mathbb{R}^p$ such that

$$h(\mathbf{x}^i) = \langle \mathbf{g}(\mathbf{x}^i; \boldsymbol{\theta}_0), \boldsymbol{\theta}^* - \boldsymbol{\theta}_0 \rangle, \quad \sqrt{m}\|\boldsymbol{\theta}^* - \boldsymbol{\theta}_0\|_2 \leq \sqrt{2\mathbf{h}^\top \mathbf{H}^{-1}\mathbf{h}}, \tag{6.1}$$

for all $i \in [TK]$.

Lemma 6.1 suggests that with high probability, the reward function restricted on $\{\mathbf{x}^i\}_{i=1}^{TK}$ can be regarded as a linear function of $\mathbf{g}(\mathbf{x}^i; \boldsymbol{\theta}_0)$ parameterized by $\boldsymbol{\theta}^* - \boldsymbol{\theta}_0$, where $\boldsymbol{\theta}^*$ lies in a ball centered at $\boldsymbol{\theta}_0$. Note that here $\boldsymbol{\theta}^*$ is not a ground truth parameter for the reward function. Instead, it is introduced only for the sake of analysis. Equipped with Lemma 6.1, we can utilize existing results on linear bandits (Abbasi-Yadkori et al., 2011) to show that with high probability, $\boldsymbol{\theta}^*$ lies in the sequence of confidence sets.

**Lemma 6.2.** There exist constants $\{\bar{C}_i\}_{i=1}^2 > 0$ such that for any $\delta \in (0, 1)$, if $\eta \leq \bar{C}_1(TmL + m\lambda)^{-1}$ and $m \geq \bar{C}_2 \max\left\{T^7\lambda^{-7}L^{21}(\log m)^3, \lambda^{-1/2}L^{-3/2}(\log(TKL^2/\delta))^{3/2}\right\}$, then with probability at least $1 - \delta$ over the random initialization of $\boldsymbol{\theta}_0$, we have $\|\boldsymbol{\theta}_t - \boldsymbol{\theta}_0\|_2 \leq 2\sqrt{t/(m\lambda)}$ and $\|\boldsymbol{\theta}^* - \boldsymbol{\theta}_t\|_{\mathbf{z}_t} \leq \gamma_t/\sqrt{m}$ for all $t \in [T]$.

**Lemma 6.3.** Denote $a_t^* = \text{argmax}_{a \in [K]} h(\mathbf{x}_{t,a})$. There exist constants $\{\bar{C}_i\}_{i=1}^3 > 0$ such that for any $\delta \in (0,1)$, if $\eta \leq \bar{C}_1(TmL + m\lambda)^{-1}, m \geq \bar{C}_2 \max\{T^7\lambda^{-7}L^{21}(\log m)^3, \lambda^{-1/2}L^{-3/2}(\log(TKL^2/\delta))^{3/2}\}$, then with probability at least $1 - \delta$ over the random initialization of $\boldsymbol{\theta}_0$, we have

$$h(\mathbf{x}_{t,a_t^*}) - h(\mathbf{x}_{t,a_t})$$
$$\leq 2\gamma_{t-1} \min\left\{\|\mathbf{g}(\mathbf{x}_{t,a_t}; \boldsymbol{\theta}_{t-1})/\sqrt{m}\|_{\mathbf{Z}_{t-1}^{-1}}, 1\right\}$$
$$+ \bar{C}_3\left(Sm^{-1/6}\sqrt{\log m}T^{7/6}\lambda^{-1/6}L^{7/2} + m^{-1/6}\sqrt{\log m}T^{5/3}\lambda^{-2/3}L^3\right),$$

Lemma 6.3 gives upper bounds for $h(\mathbf{x}_{t,a_t^*}) - h(\mathbf{x}_{t,a_t})$, which helps us to bound the regret $R_T$. It is worth noting that $\gamma_t$ has a term $\log \det \tilde{\mathbf{Z}}_t$. A trivial upper bound of $\log \det \mathbf{Z}_t$ would result in an dependence on the neural network width $m$, since the dimension of $\mathbf{Z}_t$ is $p = md + m^2(L-2) + m$. The next lemma establishes an upper bound which is independent of $m$, and is only related to effective dimension $\tilde{d}$. The dependence on $\tilde{d}$ is similar to Lemma 4 of Valko et al. (2013), but the proof is different as we use a different notion of effective dimension.

**Lemma 6.4.** There exist constants $\{\bar{C}_i\}_{i=1}^3 > 0$ such that for any $\delta \in (0,1)$, if $\eta \leq \bar{C}_1(TmL + m\lambda)^{-1}, m \geq \bar{C}_2 \max\{T^7\lambda^{-7}L^{21}(\log m)^3, T^6K^6L^6(\log(TKL^2/\delta))^{3/2}\}$, then with probability at least $1 - \delta$ over the random initialization of $\boldsymbol{\theta}_0$, we have

$$\sqrt{\sum_{t=1}^T \gamma_{t-1}^2 \min\left\{\|\mathbf{g}(\mathbf{x}_{t,a_t}; \boldsymbol{\theta}_{t-1})/\sqrt{m}\|_{\mathbf{Z}_{t-1}^{-1}}^2, 1\right\}}$$
$$\leq \sqrt{\tilde{d}\log(1 + TK/\lambda) + 1 + \bar{C}_3 m^{-1/6}\sqrt{\log m}L^4 T^{5/3}\lambda^{-1/6}} \left[\sqrt{1 + \bar{C}_3 m^{-1/6}\sqrt{\log m}L^4 T^{7/6}\lambda^{-7/6}}\right.$$
$$\cdot \left(\nu\sqrt{\tilde{d}\log(1 + TK/\lambda) + 1 + \bar{C}_3 m^{-1/6}\sqrt{\log m}L^4 T^{5/3}\lambda^{-1/6} - 2\log\delta} + \sqrt{\lambda}S\right)$$
$$\left. + \bar{C}_3\left[(1 - \eta m\lambda)^J\sqrt{T/\lambda} + m^{-1/6}\sqrt{\log m}L^{7/2}T^{5/3}\lambda^{-5/3}(1 + \sqrt{T/\lambda})\right]\right].$$

With above lemmas, we begin to prove Theorem 5.5.

*Proof of Theorem 5.5.* The total regret $R_T$ can be bounded as follows:

$$R_T = \sum_{t=1}^T \left[h(\mathbf{x}_{t,a_t^*}) - h(\mathbf{x}_{t,a_t})\right]$$
$$\leq 2\sum_{t=1}^T \gamma_{t-1} \min\left\{\|\mathbf{g}(\mathbf{x}_{t,a_t}; \boldsymbol{\theta}_{t-1})/\sqrt{m}\|_{\mathbf{Z}_{t-1}^{-1}}, 1\right\}$$
$$+ C_1\left(Sm^{-1/6}\sqrt{\log m}T^{13/6}\lambda^{-1/6}L^{7/2} + m^{-1/6}\sqrt{\log m}T^{8/3}\lambda^{-2/3}L^3\right),$$
$$\leq 2\sqrt{T\sum_{t=1}^T \gamma_{t-1}^2 \min\left\{\|\mathbf{g}(\mathbf{x}_{t,a_t}; \boldsymbol{\theta}_{t-1})/\sqrt{m}\|_{\mathbf{Z}_{t-1}^{-1}}^2, 1\right\}}$$
$$+ C_1\left(Sm^{-1/6}\sqrt{\log m}T^{13/6}\lambda^{-1/6}L^{7/2} + m^{-1/6}\sqrt{\log m}T^{8/3}\lambda^{-2/3}L^3\right)$$
$$\leq 3\sqrt{T}\sqrt{\tilde{d}\log(1 + TK/\lambda) + 2}\left[\nu\sqrt{\tilde{d}\log(1 + TK/\lambda) + 2 - 2\log\delta}\right.$$
$$\left. + 2\sqrt{\lambda}S + C_2(1 - \eta m\lambda)^J\sqrt{T/\lambda}\right] + 1,$$

where $C_1, C_2 > 0$ are constants, the first inequality holds due to Lemma 6.3, the second inequality holds Cauchy-Schwarz inequality, the third inequality holds due to Lemma 6.4 and choosing large enough $m$. This completes our proof. $\quad\square$

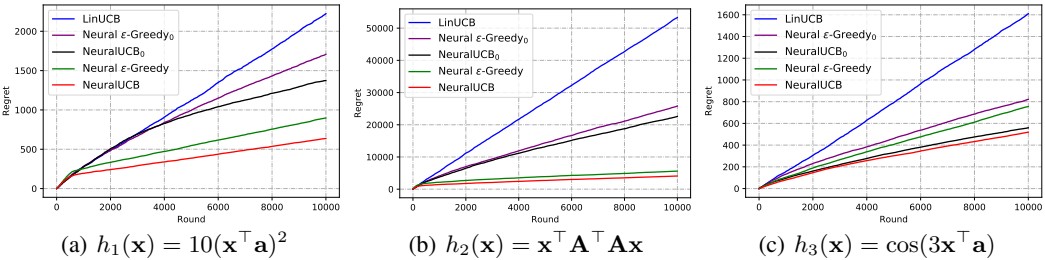

Figure 1: Comparison of LinUCB, Neural $\epsilon$-Greedy and NeuralUCB.

## 7 EXPERIMENTS

While the focus of this work is mostly on theoretical analysis of regret, we present results in proof-of-concept experiments in simulated problems. We compare it with four representative baselines: (1) LinUCB; (2) Neural $\epsilon$-Greedy, which replaces the UCB based exploration in Algorithm 1 by $\epsilon$-greedy based exploration; (3) NeuralUCB$_0$; and (4) Neural $\epsilon$-Greedy$_0$ which replaces the UCB based exploration in NeuralUCB$_0$ by $\epsilon$-greedy based exploration. We use the cumulative regret as the performance metric.

In our simulation, we use contextual bandit problems with context dimension $d = 20$, the number of actions $K = 4$ and the number of rounds $T = 10\,000$. The contextual vectors $\{\mathbf{x}_{1,1}, \ldots, \mathbf{x}_{T,K}\}$ are randomly chosen from $N(\mathbf{0}, \mathbf{I})$ and then normalized to have unit norm, i.e., $\|\mathbf{x}_{t,a}\|_2 = 1$. For the reward function $h$, we investigate the following nonlinear functions:

$$h_1(\mathbf{x}) = 10(\mathbf{x}^\top \mathbf{a})^2, \quad h_2(\mathbf{x}) = \mathbf{x}^\top \mathbf{A}^\top \mathbf{A} \mathbf{x}, \quad h_3(\mathbf{x}) = \cos(3\mathbf{x}^\top \mathbf{a}),$$

where $\mathbf{A} \in \mathbb{R}^{d \times d}$ and each entry of $\mathbf{A}$ is randomly generated from $N(0,1)$, $\mathbf{a}$ is randomly chosen from $N(\mathbf{0}, \mathbf{I})$ and normalized to have $\|\mathbf{a}\|_2 = 1$. For each $h_i(\cdot)$, the reward at round $t$ for action $a$ is generated by $r_{t,a} = h_i(\mathbf{x}_{t,a}) + \xi_t$, where $\xi_t$ is independently drawn from $N(0,1)$.

For LinUCB, we follow Li et al. (2010) to implement it with a constant radius $\alpha$. We do a grid search for $\alpha$ over $\{0.01, 0.1, 1, 10\}$ and choose the best $\alpha$ for comparison. For NeuralUCB NeuralUCB$_0$, Neural $\epsilon$-Greedy and Neural $\epsilon$-Greedy$_0$, we choose a two-layer neural network $f(\mathbf{x}; \boldsymbol{\theta}) = \sqrt{m} \mathbf{W}_2 \sigma(\mathbf{W}_1 \mathbf{x})$ with network width $m = 20$, where $\boldsymbol{\theta} = [\text{vec}(\mathbf{W}_1)^\top, \text{vec}(\mathbf{W}_2)^\top] \in \mathbb{R}^p$ and $p = md + m$. We remark that the bound on the required network width $m$ is likely not tight. Therefore, in experiments we choose $m$ to be relatively large (but not as large as theory suggests). For $\gamma_t$ in NeuralUCB, we choose $\gamma_t = \gamma$ in the experiment. We do a grid search over $\{0.01, 0.1, 1, 10\}$ and choose the best $\gamma$ for comparison. For NeuralUCB$_0$, we choose $\nu = 1, \lambda = 1, \delta = 0.1$, and we do a grid search over $\{0.01, 0.1, 1, 10\}$ for hyper-parameter $S$ to choose the best $S$ for comparison. For Neural $\epsilon$-Greedy and Neural $\epsilon$-Greedy$_0$, we do a grid search for $\epsilon$ over $\{0.001, 0.01, 0.1, 0.2\}$ and choose the best $\epsilon$ for comparison. For all the algorithms, we repeat the experiment for 10 runs and report the averaged results for comparison. For TrainNN, we choose the step size $\eta = 0.1$. To accelerate the training process, we update the parameter $\boldsymbol{\theta}_t$ by TrainNN every 50 rounds. We use stochastic gradient descent with batch size 50 and set $J = t$ at $t$-th round of NeuralUCB and Neural $\epsilon$-Greedy.

We plot the cumulative regret of compared algorithms in Figure 1, for reward function $h \in \{h_1, h_2, h_3\}$. We can see that due to the nonlinearity of reward function $h$, LinUCB fails to learn the true reward function and hence achieve an almost linear regret, as expected. In contrast, thanks to the neural network representation and efficient exploration, NeuralUCB achieves a sublinear regret which is much lower than that of LinUCB. The performance of Neural $\epsilon$-Greedy is in-between. This suggests that while Neural $\epsilon$-greedy can capture the nonlinearity of the underlying reward function, $\epsilon$-Greedy based exploration is not as effective as UCB based exploration. This confirms the effectiveness of NeuralUCB for contextual bandit problems with any bounded (nonlinear) reward function. Meanwhile, it is worth noting that NeuralUCB and Neural $\epsilon$-Greedy outperform NeuralUCB$_0$ and Neural $\epsilon$-Greedy$_0$, which suggests that using deep neural networks to predict the reward function is better than using a fixed feature mapping associated with NTK. Another observation is that although the network width $m$ in the experiment is not as large as our theory suggests, NeuralUCB still

achieves a sublinear regret for nonlinear reward functions. We leave it as a future work to investigate the impact of $m$ on regret.

## 8 CONCLUSIONS AND FUTURE WORK

In this work, we proposed a new algorithm NeuralUCB for stochastic contextual bandit problems based on neural networks. We show that for arbitrary bounded reward function, our algorithm achieves $\widetilde{O}(\widetilde{d}\sqrt{T})$ regret bound. Our preliminary experiment results on synthetic data corroborate our theoretical findings. In the future, we are interested in a systematic empirical evaluation of NeuralUCB on real world datasets, and compare it with the state-of-the-art neural network based contextual bandit algorithms (without provable guarantee in regret) (Riquelme et al., 2018; Zahavy and Mannor, 2019). Another interesting direction is provably efficient exploration with neural network using other strategies like Thompson sampling.

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

# A PROOF OF ADDITIONAL RESULTS IN SECTION 5

## A.1 VERIFICATION OF REMARK 5.4

Suppose there exists a mapping $\boldsymbol{\psi} : \mathbb{R}^d \to \mathbb{R}^{\widehat{d}}$ satisfying $\|\boldsymbol{\psi}(\mathbf{x})\|_2 \leq 1$ which maps any context $\mathbf{x} \in \mathbb{R}^d$ to the Hilbert space $\mathcal{H}$ associated with the Gram matrix $\mathbf{H} \in \mathbb{R}^{TK \times TK}$ over contexts $\{\mathbf{x}^i\}_{i=1}^{TK}$. Then $\mathbf{H} = \boldsymbol{\Psi}^\top \boldsymbol{\Psi}$, where $\boldsymbol{\Psi} = [\boldsymbol{\psi}(\mathbf{x}^1), \ldots, \boldsymbol{\psi}(\mathbf{x}^{TK})] \in \mathbb{R}^{\widehat{d} \times TK}$. Thus, we can bound the effective dimension $\widetilde{d}$ as follows

$$\widetilde{d} = \frac{\log \det[\mathbf{I} + \mathbf{H}/\lambda]}{\log(1 + TK/\lambda)} = \frac{\log \det \left[\mathbf{I} + \boldsymbol{\Psi}\boldsymbol{\Psi}^\top/\lambda\right]}{\log(1 + TK/\lambda)} \leq \widehat{d} \cdot \frac{\log \left\|\mathbf{I} + \boldsymbol{\Psi}\boldsymbol{\Psi}^\top/\lambda\right\|_2}{\log(1 + TK/\lambda)}.$$

where the second equality holds due to the fact that $\det(\mathbf{I} + \mathbf{A}^\top \mathbf{A}/\lambda) = \det(\mathbf{I} + \mathbf{A}\mathbf{A}^\top/\lambda)$ holds for any matrix $\mathbf{A}$, and the inequality holds since $\det \mathbf{A} \leq \|\mathbf{A}\|_2^{\widehat{d}}$ for any $\mathbf{A} \in \mathbb{R}^{\widehat{d} \times \widehat{d}}$. Clearly, $\widetilde{d} \leq \widehat{d}$ as long as $\left\|\mathbf{I} + \boldsymbol{\Psi}\boldsymbol{\Psi}^\top/\lambda\right\|_2 \leq 1 + TK/\lambda$. Indeed,

$$\left\|\mathbf{I} + \boldsymbol{\Psi}\boldsymbol{\Psi}^\top/\lambda\right\|_2 \leq 1 + \left\|\boldsymbol{\Psi}\boldsymbol{\Psi}^\top\right\|_2/\lambda \leq 1 + \sum_{i=1}^{TK} \left\|\boldsymbol{\psi}(\mathbf{x}^i)\boldsymbol{\psi}(\mathbf{x}^i)^\top\right\|_2/\lambda \leq 1 + TK/\lambda,$$

where the first inequality is due to triangle inequality and the fact $\lambda \geq 1$, the second inequality holds due to the definition of $\boldsymbol{\Psi}$ and triangle inequality, and the last inequality is by $\|\boldsymbol{\psi}(\mathbf{x}^i)\|_2 \leq 1$ for any $1 \leq i \leq TK$.

## A.2 VERIFICATION OF REMARK 5.8

Let $K(\cdot, \cdot)$ be the NTK kernel, then for any $i, j \in [TK]$, we have $\mathbf{H}_{i,j} = K(\mathbf{x}^i, \mathbf{x}^j)$. We first prove that $\|h\|_{\mathcal{H}} = \|\boldsymbol{\beta}^*\|_2$. Since $h \in \mathcal{H}$, we know that

$$h(\mathbf{x}) = \sum_{i=1}^n \alpha_i K(\mathbf{x}_i, \mathbf{x}) = \sum_{i=1}^n \alpha_i \langle \psi(\mathbf{x}_i), \psi(\mathbf{x}) \rangle = \Big\langle \sum_{i=1}^n \alpha_i \psi(\mathbf{x}_i), \psi(\mathbf{x}) \Big\rangle,$$

for some $\{\mathbf{x}_i\}_{i=1}^n, \{\alpha_i\}_{i=1}^n$. Thus, we have $\boldsymbol{\beta}^* = \sum_{i=1}^n \alpha_i \psi(\mathbf{x}_i)$. Therefore, we have

$$\begin{aligned}
\|h\|_{\mathcal{H}}^2 &= \sum_{i,j=1}^n \alpha_i \alpha_j K(\mathbf{x}_i, \mathbf{x}_j) \\
&= \sum_{i,j=1}^n \alpha_i \alpha_j \langle \psi(\mathbf{x}_i), \psi(\mathbf{x}_j) \rangle \\
&= \Big\langle \sum_{i=1}^n \alpha_i \psi(\mathbf{x}_i), \sum_{j=1}^n \alpha_j \psi(\mathbf{x}_j) \Big\rangle \\
&= \langle \boldsymbol{\beta}^*, \boldsymbol{\beta}^* \rangle \\
&= \|\boldsymbol{\beta}^*\|_2^2,
\end{aligned}$$

which immediately suggests that $\|h\|_{\mathcal{H}} = \|\boldsymbol{\beta}^*\|_2$. Next we prove that $\|h\|_{\mathcal{H}} \geq \sqrt{\mathbf{h}^\top \mathbf{H}^{-1} \mathbf{h}}$. First, $h$ can be decomposed as $h = h_{\mathbf{H}} + h_\perp$, where $h_{\mathbf{H}}(\mathbf{x}) = \sum_{i=1}^{TK} \widetilde{\alpha}_i K(\mathbf{x}, \mathbf{x}^i)$ is the projection of $h$ to the function space spanned by $\{K(\mathbf{x}, \mathbf{x}^i)\}_{i=1}^{TK}$ and $h_\perp$ is the orthogonal part. By definition we have $h(\mathbf{x}^i) = h_{\mathbf{H}}(\mathbf{x}^i)$ for $i \in [TK]$, thus

$$\begin{aligned}
\mathbf{h} &= [h(\mathbf{x}^1), \ldots, h(\mathbf{x}^{TK})]^\top \\
&= [h_{\mathbf{H}}(\mathbf{x}^1), \ldots, h_{\mathbf{H}}(\mathbf{x}^{TK})]^\top \\
&= \left[\sum_{i=1}^{TK} \widetilde{\alpha}_i K(\mathbf{x}^1, \mathbf{x}^i), \ldots, \sum_{i=1}^{TK} \widetilde{\alpha}_i K(\mathbf{x}^{TK}, \mathbf{x}^i)\right]^\top \\
&= \mathbf{H}\widetilde{\boldsymbol{\alpha}},
\end{aligned}$$

which implies that $\widetilde{\boldsymbol{\alpha}} = \mathbf{H}^{-1}\mathbf{h}$. Thus, we have

$$\|h\|_{\mathcal{H}} \geq \|h_{\mathbf{H}}\|_{\mathcal{H}} = \sqrt{\widetilde{\boldsymbol{\alpha}}^\top \mathbf{H}\widetilde{\boldsymbol{\alpha}}} = \sqrt{\mathbf{h}^\top \mathbf{H}^{-1} \mathbf{H} \mathbf{H}^{-1} \mathbf{h}} = \sqrt{\mathbf{h}^\top \mathbf{H}^{-1} \mathbf{h}}.$$

### A.3 PROOF OF COROLLARY 5.9

*Proof of Corollary 5.9.* Notice that $R_T \leq T$ since $0 \leq h(\mathbf{x}) \leq 1$. Thus, with the fact that with probability at least $1 - \delta$, (5.2) holds, we can bound $\mathbb{E}[R_T]$ as

$$\mathbb{E}[R_T] \leq (1 - \delta)\left(3\sqrt{T}\sqrt{\widetilde{d}\log(1 + TK/\lambda)} + 2\left[\nu\sqrt{\widetilde{d}\log(1 + TK/\lambda) + 2 - 2\log\delta}\right.\right.$$
$$\left.\left. + 2\sqrt{\lambda}S + C_2(1 - \eta m\lambda)^J\sqrt{T/\lambda}\right] + 1\right) + \delta T. \tag{A.1}$$

Taking $\delta = 1/T$ completes the proof. $\square$

## B PROOF OF LEMMAS IN SECTION 6

### B.1 PROOF OF LEMMA 6.1

We start with the following lemma:

**Lemma B.1.** Let $\mathbf{G} = [\mathbf{g}(\mathbf{x}^1; \boldsymbol{\theta}_0), \ldots, \mathbf{g}(\mathbf{x}^{TK}; \boldsymbol{\theta}_0)]/\sqrt{m} \in \mathbb{R}^{p \times (TK)}$. We denote the neural tangent kernel matrix $\mathbf{H}$ the same as Definition 5.1. For any $\delta \in (0, 1)$, if

$$m = \Omega\left(\frac{L^6 \log(TKL/\delta)}{\epsilon^4}\right),$$

then with probability at least $1 - \delta$ over the random initialization of $\boldsymbol{\theta}_0$, we have

$$\|\mathbf{G}^\top\mathbf{G} - \mathbf{H}\|_F \leq TK\epsilon.$$

We begin to prove Lemma 6.1.

*Proof of Lemma 6.1.* By Assumption 5.2, we know that $\lambda_0 > 0$. Then with probability at least $1 - \delta$, we have

$$\mathbf{G}^\top\mathbf{G} \succeq \mathbf{H} - \|\mathbf{G}^\top\mathbf{G} - \mathbf{H}\|_F\mathbf{I} \succeq \mathbf{H} - \lambda_0\mathbf{I}/2 \succeq \mathbf{H}/2 \succ 0, \tag{B.1}$$

where the second inequality holds due to Lemma B.1 with the choice of $m$, the third and fourth inequality holds due to $\mathbf{H} \succeq \lambda_0\mathbf{I} \succ 0$. Thus, suppose the singular value decomposition of $\mathbf{G}$ is $\mathbf{G} = \mathbf{PAQ}^\top$, $\mathbf{P} \in \mathbb{R}^{p \times TK}$, $\mathbf{A} \in \mathbb{R}^{TK \times TK}$, $\mathbf{Q} \in \mathbb{R}^{TK \times TK}$, we have $\mathbf{A} \succ 0$ and $\boldsymbol{\theta}^* = \boldsymbol{\theta}_0 + \mathbf{PA}^{-1}\mathbf{Q}^\top\mathbf{h}/\sqrt{m}$ satisfies (6.1). To validate that $\boldsymbol{\theta}^*$ satisfies (6.1), first we have

$$\mathbf{G}^\top\sqrt{m}(\boldsymbol{\theta}^* - \boldsymbol{\theta}_0) = \mathbf{QAP}^\top\mathbf{PA}^{-1}\mathbf{Q}^\top\mathbf{h} = \mathbf{h},$$

which suggests that for any $i$, $\langle \mathbf{g}(\mathbf{x}^i; \boldsymbol{\theta}_0), \boldsymbol{\theta}^* - \boldsymbol{\theta}_0 \rangle = h(\mathbf{x}^i)$. We also have

$$m\|\boldsymbol{\theta}^* - \boldsymbol{\theta}_0\|_2^2 = \mathbf{h}^\top\mathbf{QA}^{-2}\mathbf{Q}^\top\mathbf{h} = \mathbf{h}^\top(\mathbf{G}^\top\mathbf{G})^{-1}\mathbf{h} \leq 2\mathbf{h}^\top\mathbf{H}^{-1}\mathbf{h},$$

where the last inequality holds due to (B.1). Thus, our proof finishes. $\square$

### B.2 PROOF OF LEMMA 6.2

In this section we prove Lemma 6.2. For the simplicity, we denote $\bar{\mathbf{Z}}_t, \bar{\mathbf{b}}_t, \bar{\gamma}_t$ as follows:

$$\bar{\mathbf{Z}}_t = \lambda\mathbf{I} + \sum_{i=1}^t \mathbf{g}(\mathbf{x}_{i,a_i}; \boldsymbol{\theta}_0)\mathbf{g}(\mathbf{x}_{i,a_i}; \boldsymbol{\theta}_0)^\top/m,$$

$$\bar{\mathbf{b}}_t = \sum_{i=1}^t r_{i,a_i}\mathbf{g}(\mathbf{x}_{i,a_i}; \boldsymbol{\theta}_0)/\sqrt{m},$$

$$\bar{\gamma}_t = \nu\sqrt{\log\frac{\det\bar{\mathbf{Z}}_t}{\det\lambda\mathbf{I}} - 2\log\delta} + \sqrt{\lambda}S.$$

We need the following lemmas. The first lemma shows that the network parameter at each round $\boldsymbol{\theta}_t$ can be well approximated by $\boldsymbol{\theta}_0 + \bar{\mathbf{Z}}_t^{-1}\bar{\mathbf{b}}_t/\sqrt{m}$.

**Lemma B.2.** There exist constants $\{\bar{C}_i\}_{i=1}^5 > 0$ such that for any $\delta > 0$, if $\eta, m$ satisfy that for all $t \in [T]$,

$$2\sqrt{t/(m\lambda)} \geq \bar{C}_1 m^{-3/2} L^{-3/2} [\log(TKL^2/\delta)]^{3/2},$$

$$2\sqrt{t/(m\lambda)} \leq \bar{C}_2 \min\left\{L^{-6}[\log m]^{-3/2}, \left(m(\lambda\eta)^2 L^{-6} t^{-1} (\log m)^{-1}\right)^{3/8}\right\},$$

$$\eta \leq \bar{C}_3 (m\lambda + tmL)^{-1},$$

$$m^{1/6} \geq \bar{C}_4 \sqrt{\log m} L^{7/2} t^{7/6} \lambda^{-7/6} (1 + \sqrt{t/\lambda}),$$

then with probability at least $1 - \delta$ over the random initialization of $\boldsymbol{\theta}_0$, for any $t \in [T]$, we have that $\|\boldsymbol{\theta}_t - \boldsymbol{\theta}_0\|_2 \leq 2\sqrt{t/(m\lambda)}$ and

$$\|\boldsymbol{\theta}_t - \boldsymbol{\theta}_0 - \bar{\mathbf{Z}}_t^{-1}\bar{\mathbf{b}}_t/\sqrt{m}\|_2$$
$$\leq (1 - \eta m\lambda)^J \sqrt{t/(m\lambda)} + \bar{C}_5 m^{-2/3} \sqrt{\log m} L^{7/2} t^{5/3} \lambda^{-5/3} (1 + \sqrt{t/\lambda}).$$

Next lemma shows the error bounds for $\bar{\mathbf{Z}}_t$ and $\mathbf{Z}_t$.

**Lemma B.3.** There exist constants $\{\bar{C}_i\}_{i=1}^5 > 0$ such that for any $\delta > 0$, if $m$ satisfies that

$$\bar{C}_1 m^{-3/2} L^{-3/2} [\log(TKL^2/\delta)]^{3/2} \leq 2\sqrt{t/(m\lambda)} \leq \bar{C}_2 L^{-6} [\log m]^{-3/2}, \ \forall t \in [T],$$

then with probability at least $1 - \delta$ over the random initialization of $\boldsymbol{\theta}_0$, we have the following inequalities for any $t \in [T]$:

$$\|\mathbf{Z}_t\|_F \leq \bar{C}_3 tL,$$

$$\|\bar{\mathbf{Z}}_t - \mathbf{Z}_t\|_F \leq \bar{C}_4 m^{-1/6} \sqrt{\log m} L^4 t^{7/6} \lambda^{-1/6},$$

$$\left|\log\frac{\det(\bar{\mathbf{Z}}_t)}{\det(\lambda\mathbf{I})} - \log\frac{\det(\mathbf{Z}_t)}{\det(\lambda\mathbf{I})}\right| \leq \bar{C}_5 m^{-1/6} \sqrt{\log m} L^4 t^{5/3} \lambda^{-1/6}.$$

With above lemmas, we prove Lemma 6.2 as follows.

*Proof of Lemma 6.2.* By Lemma B.2 we know that $\|\boldsymbol{\theta}_t - \boldsymbol{\theta}_0\|_2 \leq 2\sqrt{t/(m\lambda)}$. By Lemma 6.1, with probability at least $1 - \delta$, there exists $\boldsymbol{\theta}^*$ such that for any $1 \leq t \leq T$,

$$h(\mathbf{x}_{t,a_t}) = \langle \mathbf{g}(\mathbf{x}_{t,a_t}; \boldsymbol{\theta}_0)/\sqrt{m}, \ \sqrt{m}(\boldsymbol{\theta}^* - \boldsymbol{\theta}_0)\rangle,$$

$$\sqrt{m}\|\boldsymbol{\theta}^* - \boldsymbol{\theta}_0\|_2 \leq \sqrt{2\mathbf{h}^\top\mathbf{H}^{-1}\mathbf{h}} \leq S,$$

where the second inequality holds due to the choice of $S$. Thus, by Theorem 2 in Abbasi-Yadkori et al. (2011), with probability at least $1 - 2\delta$, for any $1 \leq t \leq T$, $\boldsymbol{\theta}^*$ satisfies that

$$\|\sqrt{m}(\boldsymbol{\theta}^* - \boldsymbol{\theta}_0) - \bar{\mathbf{Z}}_t^{-1}\bar{\mathbf{b}}_t\|_{\bar{\mathbf{Z}}_t} \leq \bar{\gamma}_t. \tag{B.2}$$

We now prove that $\|\boldsymbol{\theta}^* - \boldsymbol{\theta}_t\|_{\mathbf{Z}_t} \leq \gamma_t/\sqrt{m}$. From the triangle inequality,

$$\|\boldsymbol{\theta}^* - \boldsymbol{\theta}_t\|_{\mathbf{Z}_t} \leq \underbrace{\|\boldsymbol{\theta}^* - \boldsymbol{\theta}_0 - \bar{\mathbf{Z}}_t^{-1}\bar{\mathbf{b}}_t/\sqrt{m}\|_{\mathbf{Z}_t}}_{I_1} + \underbrace{\|\boldsymbol{\theta}_t - \boldsymbol{\theta}_0 - \bar{\mathbf{Z}}_t^{-1}\bar{\mathbf{b}}_t/\sqrt{m}\|_{\mathbf{Z}_t}}_{I_2}. \tag{B.3}$$

We bound $I_1$ and $I_2$ separately. For $I_1$, we have

$$I_1^2 = (\boldsymbol{\theta}^* - \boldsymbol{\theta}_0 - \bar{\mathbf{Z}}_t^{-1}\bar{\mathbf{b}}_t/\sqrt{m})^\top \mathbf{Z}_t (\boldsymbol{\theta}^* - \boldsymbol{\theta}_0 - \bar{\mathbf{Z}}_t^{-1}\bar{\mathbf{b}}_t/\sqrt{m})$$

$$= (\boldsymbol{\theta}^* - \boldsymbol{\theta}_0 - \bar{\mathbf{Z}}_t^{-1}\bar{\mathbf{b}}_t/\sqrt{m})^\top \bar{\mathbf{Z}}_t (\boldsymbol{\theta}^* - \boldsymbol{\theta}_0 - \bar{\mathbf{Z}}_t^{-1}\bar{\mathbf{b}}_t/\sqrt{m})$$

$$+ (\boldsymbol{\theta}^* - \boldsymbol{\theta}_0 - \bar{\mathbf{Z}}_t^{-1}\bar{\mathbf{b}}_t/\sqrt{m})^\top (\mathbf{Z}_t - \bar{\mathbf{Z}}_t)(\boldsymbol{\theta}^* - \boldsymbol{\theta}_0 - \bar{\mathbf{Z}}_t^{-1}\bar{\mathbf{b}}_t/\sqrt{m})$$

$$\leq (\boldsymbol{\theta}^* - \boldsymbol{\theta}_0 - \bar{\mathbf{Z}}_t^{-1}\bar{\mathbf{b}}_t/\sqrt{m})^\top \bar{\mathbf{Z}}_t (\boldsymbol{\theta}^* - \boldsymbol{\theta}_0 - \bar{\mathbf{Z}}_t^{-1}\bar{\mathbf{b}}_t/\sqrt{m})$$

$$+ \frac{\|\mathbf{Z}_t - \bar{\mathbf{Z}}_t\|_2}{\lambda}(\boldsymbol{\theta}^* - \boldsymbol{\theta}_0 - \bar{\mathbf{Z}}_t^{-1}\bar{\mathbf{b}}_t/\sqrt{m})^\top \bar{\mathbf{Z}}_t (\boldsymbol{\theta}^* - \boldsymbol{\theta}_0 - \bar{\mathbf{Z}}_t^{-1}\bar{\mathbf{b}}_t/\sqrt{m})$$

$$\leq (1 + \|\mathbf{Z}_t - \bar{\mathbf{Z}}_t\|_2/\lambda)\bar{\gamma}_t^2/m, \tag{B.4}$$

where the first inequality holds due to the fact that $\mathbf{x}^\top \mathbf{A}\mathbf{x} \leq \mathbf{x}^\top \mathbf{B}\mathbf{x} \cdot \|\mathbf{A}\|_2 / \lambda_{\min}(\mathbf{B})$ for some $\mathbf{B} \succ 0$ and the fact that $\lambda_{\min}(\bar{\mathbf{Z}}_t) \geq \lambda$, the second inequality holds due to (B.2). By Lemma B.3, we have

$$\left\|\bar{\mathbf{Z}}_t - \mathbf{Z}_t\right\|_2 \leq \left\|\bar{\mathbf{Z}}_t - \mathbf{Z}_t\right\|_F \leq C_1 m^{-1/6} \sqrt{\log m} L^4 t^{7/6} \lambda^{-1/6}, \tag{B.5}$$

and

$$\begin{aligned}
\bar{\gamma}_t &= \nu \sqrt{\log \frac{\det \bar{\mathbf{Z}}_t}{\det \lambda \mathbf{I}} - 2\log \delta} + \sqrt{\lambda} S \\
&\leq \nu \sqrt{\log \frac{\det \mathbf{Z}_t}{\det \lambda \mathbf{I}} + C_2 m^{-1/6} \sqrt{\log m} L^4 t^{5/3} \lambda^{-1/6} - 2\log \delta} + \sqrt{\lambda} S,
\end{aligned} \tag{B.6}$$

where $C_1, C_2 > 0$ are two constants. Substituting (B.5) and (B.6) into (B.4), we have

$$\begin{aligned}
I_1 &\leq \sqrt{1 + \|\mathbf{Z}_t - \bar{\mathbf{Z}}_t\|_2 / \lambda} \bar{\gamma}_t / \sqrt{m} \\
&\leq \sqrt{1 + C_1 m^{-1/6} \sqrt{\log m} L^4 t^{7/6} \lambda^{-7/6}} / \sqrt{m} \\
&\quad \cdot \left( \nu \sqrt{\log \frac{\det \mathbf{Z}_t}{\det \lambda \mathbf{I}} + C_2 m^{-1/6} \sqrt{\log m} L^4 t^{5/3} \lambda^{-1/6} - 2\log \delta} + \sqrt{\lambda} S \right).
\end{aligned} \tag{B.7}$$

For $I_2$, we have

$$\begin{aligned}
I_2 &\leq C_3 tL \|\boldsymbol{\theta}_t - \boldsymbol{\theta}_0 - \bar{\mathbf{Z}}_t^{-1} \bar{\mathbf{b}}_t / \sqrt{m}\|_2 \\
&\leq C_4 \left[ (1 - \eta m\lambda)^J \sqrt{t/(m\lambda)} + m^{-2/3} \sqrt{\log m} L^{7/2} t^{5/3} \lambda^{-5/3} (1 + \sqrt{t/\lambda}) \right],
\end{aligned} \tag{B.8}$$

where $C_3, C_4 > 0$ are two constants, the first inequality holds due to $\|\mathbf{Z}_t\|_2 \leq \|\mathbf{Z}_t\|_F \leq C_3 tL$ by Lemma B.3, the second inequality holds due to Lemma B.2. Substituting (B.7) and (B.8) into (B.3), we have $\left\|\boldsymbol{\theta}^* - \boldsymbol{\theta}_t\right\|_{\mathbf{Z}_t} \leq \gamma_t / \sqrt{m}$. Our proof is thus completed. $\qquad \square$

### B.3 PROOF OF LEMMA 6.3

The proof starts with three lemmas that bound the error terms of the function value and gradient of neural networks.

**Lemma B.4** (Lemma 4.1, Cao and Gu (2019a)). There exist constants $\{\bar{C}_i\}_{i=1}^3 > 0$ such that for any $\delta > 0$, if $\tau$ satisfies that

$$\bar{C}_1 m^{-3/2} L^{-3/2} [\log(TKL^2/\delta)]^{3/2} \leq \tau \leq \bar{C}_2 L^{-6} [\log m]^{-3/2},$$

then with probability at least $1 - \delta$ over the random initialization of $\boldsymbol{\theta}_0$, for all $\widetilde{\boldsymbol{\theta}}, \widehat{\boldsymbol{\theta}}$ satisfying $\|\widetilde{\boldsymbol{\theta}} - \boldsymbol{\theta}_0\|_2 \leq \tau, \|\widehat{\boldsymbol{\theta}} - \boldsymbol{\theta}_0\|_2 \leq \tau$ and $j \in [TK]$ we have

$$\left| f(\mathbf{x}^j; \widetilde{\boldsymbol{\theta}}) - f(\mathbf{x}^j; \widehat{\boldsymbol{\theta}}) - \langle \mathbf{g}(\mathbf{x}^j; \widehat{\boldsymbol{\theta}}), \widetilde{\boldsymbol{\theta}} - \widehat{\boldsymbol{\theta}} \rangle \right| \leq \bar{C}_3 \tau^{4/3} L^3 \sqrt{m \log m}.$$

**Lemma B.5** (Theorem 5, Allen-Zhu et al. (2019)). There exist constants $\{\bar{C}_i\}_{i=1}^3 > 0$ such that for any $\delta \in (0, 1)$, if $\tau$ satisfies that

$$\bar{C}_1 m^{-3/2} L^{-3/2} [\log(TKL^2/\delta)]^{3/2} \leq \tau \leq \bar{C}_2 L^{-6} [\log m]^{-3/2},$$

then with probability at least $1 - \delta$ over the random initialization of $\boldsymbol{\theta}_0$, for all $\|\boldsymbol{\theta} - \boldsymbol{\theta}_0\|_2 \leq \tau$ and $j \in [TK]$ we have

$$\|\mathbf{g}(\mathbf{x}^j; \boldsymbol{\theta}) - \mathbf{g}(\mathbf{x}^j; \boldsymbol{\theta}_0)\|_2 \leq \bar{C}_3 \sqrt{\log m} \tau^{1/3} L^3 \|\mathbf{g}(\mathbf{x}^j; \boldsymbol{\theta}_0)\|_2.$$

**Lemma B.6** (Lemma B.3, Cao and Gu (2019a)). There exist constants $\{\bar{C}_i\}_{i=1}^3 > 0$ such that for any $\delta > 0$, if $\tau$ satisfies that

$$\bar{C}_1 m^{-3/2} L^{-3/2} [\log(TKL^2/\delta)]^{3/2} \leq \tau \leq \bar{C}_2 L^{-6} [\log m]^{-3/2},$$

then with probability at least $1 - \delta$ over the random initialization of $\boldsymbol{\theta}_0$, for any $\|\boldsymbol{\theta} - \boldsymbol{\theta}_0\|_2 \leq \tau$ and $j \in [TK]$ we have $\|\mathbf{g}(\mathbf{x}^j; \boldsymbol{\theta})\|_F \leq \bar{C}_3 \sqrt{m L}$.

*Proof of Lemma 6.3.* We follow the regret bound analysis in Abbasi-Yadkori et al. (2011); Valko et al. (2013). Denote $a_t^* = \operatorname{argmax}_{a \in [K]} h(\mathbf{x}_{t,a})$ and $\mathcal{C}_t = \{\boldsymbol{\theta} : \|\boldsymbol{\theta} - \boldsymbol{\theta}_t\|_{\mathbf{Z}_t} \leq \gamma_t/\sqrt{m}\}$. By Lemma 6.2, for all $1 \leq t \leq T$, we have $\|\boldsymbol{\theta}_t - \boldsymbol{\theta}_0\|_2 \leq 2\sqrt{t/(m\lambda)}$ and $\boldsymbol{\theta}^* \in \mathcal{C}_t$. By the choice of $m$, Lemma B.4, B.5 and B.6 hold. Thus, $h(\mathbf{x}_{t,a_t^*}) - h(\mathbf{x}_{t,a_t})$ can be bounded as follows:

$$
\begin{aligned}
&h(\mathbf{x}_{t,a_t^*}) - h(\mathbf{x}_{t,a_t}) \\
&= \langle \mathbf{g}(\mathbf{x}_{t,a_t^*}; \boldsymbol{\theta}_0), \boldsymbol{\theta}^* - \boldsymbol{\theta}_0 \rangle - \langle \mathbf{g}(\mathbf{x}_{t,a_t}; \boldsymbol{\theta}_0), \boldsymbol{\theta}^* - \boldsymbol{\theta}_0 \rangle \\
&\leq \langle \mathbf{g}(\mathbf{x}_{t,a_t^*}; \boldsymbol{\theta}_{t-1}), \boldsymbol{\theta}^* - \boldsymbol{\theta}_0 \rangle - \langle \mathbf{g}(\mathbf{x}_{t,a_t}; \boldsymbol{\theta}_{t-1}), \boldsymbol{\theta}^* - \boldsymbol{\theta}_0 \rangle \\
&\quad + \|\boldsymbol{\theta}^* - \boldsymbol{\theta}_0\|_2 (\|\mathbf{g}(\mathbf{x}_{t,a_t^*}; \boldsymbol{\theta}_{t-1}) - \mathbf{g}(\mathbf{x}_{t,a_t^*}; \boldsymbol{\theta}_0)\|_2 + \|\mathbf{g}(\mathbf{x}_{t,a_t}; \boldsymbol{\theta}_{t-1}) - \mathbf{g}(\mathbf{x}_{t,a_t}; \boldsymbol{\theta}_0)\|_2) \\
&\leq \langle \mathbf{g}(\mathbf{x}_{t,a_t^*}; \boldsymbol{\theta}_{t-1}), \boldsymbol{\theta}^* - \boldsymbol{\theta}_0 \rangle - \langle \mathbf{g}(\mathbf{x}_{t,a_t}; \boldsymbol{\theta}_{t-1}), \boldsymbol{\theta}^* - \boldsymbol{\theta}_0 \rangle + C_1 \sqrt{\mathbf{h}^\top \mathbf{H}^{-1} \mathbf{h}} m^{-1/6} \sqrt{\log m} t^{1/6} \lambda^{-1/6} L^{7/2} \\
&\leq \max_{\boldsymbol{\theta} \in \mathcal{C}_{t-1}} \langle \mathbf{g}(\mathbf{x}_{t,a_t^*}; \boldsymbol{\theta}_{t-1}), \boldsymbol{\theta} - \boldsymbol{\theta}_0 \rangle - \langle \mathbf{g}(\mathbf{x}_{t,a_t}; \boldsymbol{\theta}_{t-1}), \boldsymbol{\theta}^* - \boldsymbol{\theta}_0 \rangle \\
&\quad + C_1 \sqrt{\mathbf{h}^\top \mathbf{H}^{-1} \mathbf{h}} m^{-1/6} \sqrt{\log m} t^{1/6} \lambda^{-1/6} L^{7/2},
\end{aligned}
\tag{B.9}
$$

where the first inequality holds due to triangle inequality, the second inequality holds due to Lemma 6.1, Lemma B.5 and B.6, the third inequality holds due to $\boldsymbol{\theta}^* \in \mathcal{C}_{t-1}$. Denote

$$
\widetilde{U}_{t,a} = \langle \mathbf{g}(\mathbf{x}_{t,a}; \boldsymbol{\theta}_{t-1}), \boldsymbol{\theta}_{t-1} - \boldsymbol{\theta}_0 \rangle + \gamma_{t-1} \sqrt{\mathbf{g}(\mathbf{x}_{t,a}; \boldsymbol{\theta}_{t-1})^\top \mathbf{Z}_{t-1}^{-1} \mathbf{g}(\mathbf{x}_{t,a}; \boldsymbol{\theta}_{t-1})/m},
$$

then we have $\widetilde{U}_{t,a} = \max_{\boldsymbol{\theta} \in \mathcal{C}_{t-1}} \langle \mathbf{g}(\mathbf{x}_{t,a}; \boldsymbol{\theta}_{t-1}), \boldsymbol{\theta} - \boldsymbol{\theta}_0 \rangle$ with the fact that

$$
\langle \mathbf{a}, \mathbf{b} \rangle + c \sqrt{\mathbf{a}^\top \mathbf{A}^{-1} \mathbf{a}} = \max_{\|\mathbf{x} - \mathbf{b}\|_{\mathbf{A}} \leq c} \langle \mathbf{a}, \mathbf{x} \rangle.
$$

We also have

$$
|U_{t,a} - \widetilde{U}_{t,a}| \leq C_2 m^{-1/6} \sqrt{\log m} t^{2/3} \lambda^{-2/3} L^3,
\tag{B.10}
$$

where $C_2 > 0$ is a constant, the inequality holds due to Lemma B.4 with the fact $\|\boldsymbol{\theta}_{t-1} - \boldsymbol{\theta}_0\|_2 \leq 2\sqrt{t/(m\lambda)}$ and the fact $f(\mathbf{x}^j; \boldsymbol{\theta}_0) = 0$. Since $\boldsymbol{\theta}^* \in \mathcal{C}_{t-1}$, then (B.9) can be bounded as

$$
\begin{aligned}
&\max_{\boldsymbol{\theta} \in \mathcal{C}_{t-1}} \langle \mathbf{g}(\mathbf{x}_{t,a_t^*}; \boldsymbol{\theta}_{t-1}), \boldsymbol{\theta} - \boldsymbol{\theta}_0 \rangle - \langle \mathbf{g}(\mathbf{x}_{t,a_t}; \boldsymbol{\theta}_{t-1}), \boldsymbol{\theta}^* - \boldsymbol{\theta}_0 \rangle \\
&= \widetilde{U}_{t,a_t^*} - \langle \mathbf{g}(\mathbf{x}_{t,a_t}; \boldsymbol{\theta}_{t-1}), \boldsymbol{\theta}^* - \boldsymbol{\theta}_0 \rangle \\
&\leq U_{t,a_t^*} - \langle \mathbf{g}(\mathbf{x}_{t,a_t}; \boldsymbol{\theta}_{t-1}), \boldsymbol{\theta}^* - \boldsymbol{\theta}_0 \rangle + C_2 m^{-1/6} \sqrt{\log m} t^{2/3} \lambda^{-2/3} L^3 \\
&\leq U_{t,a_t} - \langle \mathbf{g}(\mathbf{x}_{t,a_t}; \boldsymbol{\theta}_{t-1}), \boldsymbol{\theta}^* - \boldsymbol{\theta}_0 \rangle + C_2 m^{-1/6} \sqrt{\log m} t^{2/3} \lambda^{-2/3} L^3 \\
&\leq \widetilde{U}_{t,a_t} - \langle \mathbf{g}(\mathbf{x}_{t,a_t}; \boldsymbol{\theta}_{t-1}), \boldsymbol{\theta}^* - \boldsymbol{\theta}_0 \rangle + 2C_2 m^{-1/6} \sqrt{\log m} t^{2/3} \lambda^{-2/3} L^3,
\end{aligned}
\tag{B.11}
$$

where the first inequality holds due to (B.10), the second inequality holds since $a_t = \operatorname{argmax}_a U_{t,a}$, the third inequality holds due to (B.10). Furthermore,

$$
\begin{aligned}
&\widetilde{U}_{t,a_t} - \langle \mathbf{g}(\mathbf{x}_{t,a_t}; \boldsymbol{\theta}_0), \boldsymbol{\theta}^* - \boldsymbol{\theta}_0 \rangle \\
&= \max_{\boldsymbol{\theta} \in \mathcal{C}_{t-1}} \langle \mathbf{g}(\mathbf{x}_{t,a_t}; \boldsymbol{\theta}_{t-1}), \boldsymbol{\theta} - \boldsymbol{\theta}_0 \rangle - \langle \mathbf{g}(\mathbf{x}_{t,a_t}; \boldsymbol{\theta}_{t-1}), \boldsymbol{\theta}^* - \boldsymbol{\theta}_0 \rangle \\
&= \max_{\boldsymbol{\theta} \in \mathcal{C}_{t-1}} \langle \mathbf{g}(\mathbf{x}_{t,a_t}; \boldsymbol{\theta}_{t-1}), \boldsymbol{\theta} - \boldsymbol{\theta}_{t-1} \rangle - \langle \mathbf{g}(\mathbf{x}_{t,a_t}; \boldsymbol{\theta}_{t-1}), \boldsymbol{\theta}^* - \boldsymbol{\theta}_{t-1} \rangle \\
&\leq \max_{\boldsymbol{\theta} \in \mathcal{C}_{t-1}} \|\boldsymbol{\theta} - \boldsymbol{\theta}_{t-1}\|_{\mathbf{Z}_{t-1}} \|\mathbf{g}(\mathbf{x}_{t,a_t}; \boldsymbol{\theta}_{t-1})\|_{\mathbf{Z}_{t-1}^{-1}} + \|\boldsymbol{\theta}^* - \boldsymbol{\theta}_{t-1}\|_{\mathbf{Z}_{t-1}} \|\mathbf{g}(\mathbf{x}_{t,a_t}; \boldsymbol{\theta}_{t-1})\|_{\mathbf{Z}_{t-1}^{-1}} \\
&\leq 2\gamma_{t-1} \|\mathbf{g}(\mathbf{x}_{t,a_t}; \boldsymbol{\theta}_{t-1})/\sqrt{m}\|_{\mathbf{Z}_{t-1}^{-1}},
\end{aligned}
\tag{B.12}
$$

where the first inequality holds due to Cauchy-Schwarz inequality, the second inequality holds due to Lemma 6.2. Combining (B.9), (B.11) and (B.12), we have

$$
\begin{aligned}
&h(\mathbf{x}_{t,a_t^*}) - h(\mathbf{x}_{t,a_t}) \\
&\leq 2\gamma_{t-1}\|\mathbf{g}(\mathbf{x}_{t,a_t};\boldsymbol{\theta}_{t-1})/\sqrt{m}\|_{\mathbf{Z}_{t-1}^{-1}} + C_1\sqrt{\mathbf{h}^\top\mathbf{H}^{-1}\mathbf{h}}m^{-1/6}\sqrt{\log m}t^{1/6}\lambda^{-1/6}L^{7/2} \\
&\qquad + 2C_2 m^{-1/6}\sqrt{\log m}t^{2/3}\lambda^{-2/3}L^3 \\
&\leq \min\Big\{2\gamma_{t-1}\|\mathbf{g}(\mathbf{x}_{t,a_t};\boldsymbol{\theta}_{t-1})/\sqrt{m}\|_{\mathbf{Z}_{t-1}^{-1}} + C_1\sqrt{\mathbf{h}^\top\mathbf{H}^{-1}\mathbf{h}}m^{-1/6}\sqrt{\log m}t^{1/6}\lambda^{-1/6}L^{7/2} \\
&\qquad + 2C_2 m^{-1/6}\sqrt{\log m}t^{2/3}\lambda^{-2/3}L^3, 1\Big\} \\
&\leq \min\Big\{2\gamma_{t-1}\|\mathbf{g}(\mathbf{x}_{t,a_t};\boldsymbol{\theta}_{t-1})/\sqrt{m}\|_{\mathbf{Z}_{t-1}^{-1}}, 1\Big\} + C_1\sqrt{\mathbf{h}^\top\mathbf{H}^{-1}\mathbf{h}}m^{-1/6}\sqrt{\log m}t^{1/6}\lambda^{-1/6}L^{7/2} \\
&\qquad + 2C_2 m^{-1/6}\sqrt{\log m}t^{2/3}\lambda^{-2/3}L^3 \\
&\leq 2\gamma_{t-1}\min\Big\{\|\mathbf{g}(\mathbf{x}_{t,a_t};\boldsymbol{\theta}_{t-1})/\sqrt{m}\|_{\mathbf{Z}_{t-1}^{-1}}, 1\Big\} + C_1\sqrt{\mathbf{h}^\top\mathbf{H}^{-1}\mathbf{h}}m^{-1/6}\sqrt{\log m}t^{1/6}\lambda^{-1/6}L^{7/2} \\
&\qquad + 2C_2 m^{-1/6}\sqrt{\log m}t^{2/3}\lambda^{-2/3}L^3,
\end{aligned}
\tag{B.13}
$$

where the second inequality holds due to the fact that $0 \leq h(\mathbf{x}_{t,a_t^*}) - h(\mathbf{x}_{t,a_t}) \leq 1$, the third inequality holds due to the fact that $\min\{a+b,1\} \leq \min\{a,1\}+b$, the fourth inequality holds due to the fact $\gamma_{t-1} \geq \sqrt{\lambda}S \geq 1$. Finally, by the fact that $\sqrt{2\mathbf{h}\mathbf{H}^{-1}\mathbf{h}} \leq S$, we finish the proof. $\qquad\square$

### B.4 PROOF OF LEMMA 6.4

In this section we prove Lemma 6.4, we need the following lemma from Abbasi-Yadkori et al. (2011).

**Lemma B.7** (Lemma 11, Abbasi-Yadkori et al. (2011))**.** We have the following inequality:

$$
\sum_{t=1}^{T}\min\Big\{\|\mathbf{g}(\mathbf{x}_{t,a_t};\boldsymbol{\theta}_{t-1})/\sqrt{m}\|_{\mathbf{Z}_{t-1}^{-1}}^2, 1\Big\} \leq 2\log\frac{\det\mathbf{Z}_T}{\det\lambda\mathbf{I}}.
$$

*Proof of Lemma 6.4.* First by the definition of $\gamma_t$, we know that $\gamma_t$ is a monotonic function w.r.t. $\det\mathbf{Z}_t$. By the definition of $\mathbf{Z}_t$, we know that $\mathbf{Z}_T \succeq \mathbf{Z}_t$, which implies that $\det\mathbf{Z}_t \leq \det\mathbf{Z}_T$. Thus, $\gamma_t \leq \gamma_T$. Second, by Lemma B.7 we know that

$$
\begin{aligned}
&\sum_{t=1}^{T}\min\Big\{\|\mathbf{g}(\mathbf{x}_{t,a_t};\boldsymbol{\theta}_{t-1})/\sqrt{m}\|_{\mathbf{Z}_{t-1}^{-1}}^2, 1\Big\} \\
&\leq 2\log\frac{\det\mathbf{Z}_T}{\det\lambda\mathbf{I}} \\
&\leq 2\log\frac{\det\bar{\mathbf{Z}}_T}{\det\lambda\mathbf{I}} + C_1 m^{-1/6}\sqrt{\log m}L^4 T^{5/3}\lambda^{-1/6},
\end{aligned}
\tag{B.14}
$$

where the second inequality holds due to Lemma B.3. Next we are going to bound $\log\det\bar{\mathbf{Z}}_T$. Denote $\mathbf{G} = [\mathbf{g}(\mathbf{x}^1;\boldsymbol{\theta}_0)/\sqrt{m}, \dots, \mathbf{g}(\mathbf{x}^{TK};\boldsymbol{\theta}_0)/\sqrt{m}] \in \mathbb{R}^{p\times(TK)}$, then we have

$$
\begin{aligned}
\log\frac{\det\bar{\mathbf{Z}}_T}{\det\lambda\mathbf{I}} &= \log\det\Big(\mathbf{I} + \sum_{t=1}^{T}\mathbf{g}(\mathbf{x}_{t,a_t};\boldsymbol{\theta}_0)\mathbf{g}(\mathbf{x}_{t,a_t};\boldsymbol{\theta}_0)^\top/(m\lambda)\Big) \\
&\leq \log\det\Big(\mathbf{I} + \sum_{i=1}^{TK}\mathbf{g}(\mathbf{x}^i;\boldsymbol{\theta}_0)\mathbf{g}(\mathbf{x}^i;\boldsymbol{\theta}_0)^\top/(m\lambda)\Big) \\
&= \log\det\Big(\mathbf{I} + \mathbf{G}\mathbf{G}^\top/\lambda\Big) \\
&= \log\det\Big(\mathbf{I} + \mathbf{G}^\top\mathbf{G}/\lambda\Big),
\end{aligned}
\tag{B.15}
$$

where the inequality holds naively, the third equality holds since for any matrix $\mathbf{A} \in \mathbb{R}^{p \times TK}$, we have $\det(\mathbf{I} + \mathbf{A}\mathbf{A}^\top) = \det(\mathbf{I} + \mathbf{A}^\top\mathbf{A})$. We can can be further bound (B.15) by the follows:

$$
\begin{aligned}
\log\det\left(\mathbf{I} + \mathbf{G}^\top\mathbf{G}/\lambda\right) &= \log\det\left(\mathbf{I} + \mathbf{H}/\lambda + (\mathbf{G}^\top\mathbf{G} - \mathbf{H})/\lambda\right) \\
&\leq \log\det\left(\mathbf{I} + \mathbf{H}/\lambda\right) + \langle(\mathbf{I} + \mathbf{H}/\lambda)^{-1}, (\mathbf{G}^\top\mathbf{G} - \mathbf{H})/\lambda\rangle \\
&\leq \log\det\left(\mathbf{I} + \mathbf{H}/\lambda\right) + \|(\mathbf{I} + \mathbf{H}/\lambda)^{-1}\|_F\|\mathbf{G}^\top\mathbf{G} - \mathbf{H}\|_F/\lambda \\
&\leq \log\det\left(\mathbf{I} + \mathbf{H}/\lambda\right) + \sqrt{TK}\|\mathbf{G}^\top\mathbf{G} - \mathbf{H}\|_F \\
&\leq \log\det\left(\mathbf{I} + \mathbf{H}/\lambda\right) + 1 \\
&\leq \widetilde{d}\log(1 + TK/\lambda) + 1,
\end{aligned}
\tag{B.16}
$$

where the first inequality holds due to the concavity of $\log\det(\cdot)$, the second inequality holds due to the fact that $\langle\mathbf{A}, \mathbf{B}\rangle \leq \|\mathbf{A}\|_F\|\mathbf{B}\|_F$, the third inequality holds due to the facts that $\mathbf{I} + \mathbf{H}/\lambda \succeq \mathbf{I}$, $\lambda \geq 1$ and $\|\mathbf{A}\|_F \leq \sqrt{TK}\|\mathbf{A}\|_2$ for any $\mathbf{A} \in \mathbb{R}^{TK \times TK}$, the fourth inequality holds by Lemma B.1 with the choice of $m$, the fifth inequality holds by the definition of effective dimension in Definition 5.3, and the last inequality holds due to the choice of $\lambda$. We now bound $\gamma_T$, which is

$$
\begin{aligned}
\gamma_T &= \sqrt{1 + C_1 m^{-1/6}\sqrt{\log m}L^4 T^{7/6}\lambda^{-7/6}} \\
&\quad \cdot \left(\nu\sqrt{\log\frac{\det \mathbf{Z}_T}{\det \lambda\mathbf{I}} + C_2 m^{-1/6}\sqrt{\log m}L^4 T^{5/3}\lambda^{-1/6} - 2\log\delta} + \sqrt{\lambda}S\right) \\
&\quad + C_3\left[(1 - \eta m\lambda)^J\sqrt{T/(m\lambda)} + m^{-2/3}\sqrt{\log m}L^{7/2}T^{5/3}\lambda^{-5/3}(1 + \sqrt{T/\lambda})\right] \\
&\leq \sqrt{1 + C_1 m^{-1/6}\sqrt{\log m}L^4 T^{7/6}\lambda^{-7/6}} \\
&\quad \cdot \left(\nu\sqrt{\log\frac{\det \overline{\mathbf{Z}}_T}{\det \lambda\mathbf{I}} + C_2 m^{-1/6}\sqrt{\log m}L^4 T^{5/3}\lambda^{-1/6} - 2\log\delta} + \sqrt{\lambda}S\right) \\
&\quad + C_3\left[(1 - \eta m\lambda)^J\sqrt{T/(m\lambda)} + m^{-2/3}\sqrt{\log m}L^{7/2}T^{5/3}\lambda^{-5/3}(1 + \sqrt{T/\lambda})\right],
\end{aligned}
\tag{B.17}
$$

where the inequality holds due to Lemma B.3. Finally, substituting (B.15), (B.16) into (B.14), using (B.14) and (B.17), we have

$$
\sqrt{\sum_{t=1}^{T} \gamma_{t-1}^2 \min \left\{ \|\mathbf{g}(\mathbf{x}_{t,a_t}; \boldsymbol{\theta}_{t-1})/\sqrt{m}\|_{\mathbf{Z}_{t-1}^{-1}}^2, 1 \right\}}
$$

$$
\leq \gamma_T \sqrt{\sum_{t=1}^{T} \min \left\{ \|\mathbf{g}(\mathbf{x}_{t,a_t}; \boldsymbol{\theta}_{t-1})/\sqrt{m}\|_{\mathbf{Z}_{t-1}^{-1}}^2, 1 \right\}}
$$

$$
\leq \sqrt{\log \frac{\det \bar{\mathbf{Z}}_T}{\det \lambda \mathbf{I}} + C_1 m^{-1/6} \sqrt{\log m} L^4 T^{5/3} \lambda^{-1/6}} \left[ \sqrt{1 + C_1 m^{-1/6} \sqrt{\log m} L^4 T^{7/6} \lambda^{-7/6}} \right.
$$

$$
\cdot \left( \nu \sqrt{\log \frac{\det \bar{\mathbf{Z}}_T}{\det \lambda \mathbf{I}} + C_2 m^{-1/6} \sqrt{\log m} L^4 T^{5/3} \lambda^{-1/6} - 2\log \delta} + \sqrt{\lambda} S \right)
$$

$$
\left. + C_3 \left[ (1 - \eta m \lambda)^J \sqrt{T/(m\lambda)} + m^{-3/2} \sqrt{\log m} L^{7/2} T^{5/3} \lambda^{-5/3} (1 + \sqrt{T/\lambda}) \right] \right]
$$

$$
\leq \sqrt{\widetilde{d} \log(1 + TK/\lambda) + 1 + C_1 m^{-1/6} \sqrt{\log m} L^4 T^{5/3} \lambda^{-1/6}} \left[ \sqrt{1 + C_1 m^{-1/6} \sqrt{\log m} L^4 T^{7/6} \lambda^{-7/6}} \right.
$$

$$
\cdot \left( \nu \sqrt{\widetilde{d} \log(1 + TK/\lambda) + 1 + C_2 m^{-1/6} \sqrt{\log m} L^4 T^{5/3} \lambda^{-1/6} - 2\log \delta} + \sqrt{\lambda} S \right)
$$

$$
\left. + C_3 \left[ (1 - \eta m \lambda)^J \sqrt{T/(m\lambda)} + m^{-3/2} \sqrt{\log m} L^{7/2} T^{5/3} \lambda^{-5/3} (1 + \sqrt{T/\lambda}) \right] \right].
$$

$\square$

## C    PROOFS OF TECHNICAL LEMMAS IN APPENDIX B

### C.1    PROOF OF LEMMA B.1

In this section we prove Lemma B.1, we need the following lemma from Arora et al. (2019):

**Lemma C.1** (Theorem 3.1, Arora et al. (2019)). Fix $\epsilon > 0$ and $\delta \in (0, 1)$. Suppose that

$$
m = \Omega\left( \frac{L^6 \log(L/\delta)}{\epsilon^4} \right),
$$

then for any $i, j \in [TK]$, with probability at least $1 - \delta$ over random initialization of $\boldsymbol{\theta}_0$, we have

$$
|\langle \mathbf{g}(\mathbf{x}^i; \boldsymbol{\theta}_0), \mathbf{g}(\mathbf{x}^j; \boldsymbol{\theta}_0) \rangle / m - \mathbf{H}_{i,j}| \leq \epsilon. \tag{C.1}
$$

*Proof of Lemma B.1.* Taking union bound over $i, j \in [TK]$, we have that if

$$
m = \Omega\left( \frac{L^6 \log(T^2 K^2 L/\delta)}{\epsilon^4} \right),
$$

then with probability at least $1 - \delta$, (C.1) holds for all $(i, j) \in [TK] \times [TK]$. Therefore, we have

$$
\|\mathbf{G}^\top \mathbf{G} - \mathbf{H}\|_F = \sqrt{\sum_{i=1}^{TK} \sum_{j=1}^{TK} |\langle \mathbf{g}(\mathbf{x}^i; \boldsymbol{\theta}_0), \mathbf{g}(\mathbf{x}^j; \boldsymbol{\theta}_0) \rangle / m - \mathbf{H}_{i,j}|^2} \leq TK\epsilon.
$$

$\square$

## C.2 PROOF OF LEMMA B.2

In this section we prove Lemma B.2. During the proof, for the simplicity, we omit the subscript $t$ by default. We define the following quantities:

$$\mathbf{J}^{(j)} = \Big( \mathbf{g}(\mathbf{x}_{1,a_1}; \boldsymbol{\theta}^{(j)}), \dots, \mathbf{g}(\mathbf{x}_{t,a_t}; \boldsymbol{\theta}^{(j)}) \Big) \in \mathbb{R}^{(md+m^2(L-2)+m) \times t},$$

$$\mathbf{H}^{(j)} = [\mathbf{J}^{(j)}]^\top \mathbf{J}^{(j)} \in \mathbb{R}^{t \times t},$$

$$\mathbf{f}^{(j)} = (f(\mathbf{x}_{1,a_1}; \boldsymbol{\theta}^{(j)}), \dots, f(\mathbf{x}_{t,a_t}; \boldsymbol{\theta}^{(j)}))^\top \in \mathbb{R}^{t \times 1},$$

$$\mathbf{v} = (r_{1,a_1}, \dots, r_{t,a_t}) \in \mathbb{R}^{t \times 1}.$$

Then the update rule of $\boldsymbol{\theta}^{(j)}$ can be written as follows:

$$\boldsymbol{\theta}^{(j+1)} = \boldsymbol{\theta}^{(j)} - \eta \big[ \mathbf{J}^{(j)}(\mathbf{f}^{(j)} - \mathbf{v}) + m\lambda(\boldsymbol{\theta}^{(j)} - \boldsymbol{\theta}^{(0)}) \big]. \tag{C.2}$$

We also define the following auxiliary sequence $\{\widetilde{\boldsymbol{\theta}}^{(k)}\}$ during the proof:

$$\widetilde{\boldsymbol{\theta}}^{(0)} = \boldsymbol{\theta}^{(0)}, \ \widetilde{\boldsymbol{\theta}}^{(j+1)} = \widetilde{\boldsymbol{\theta}}^{(j)} - \eta \big[ \mathbf{J}^{(0)}([\mathbf{J}^{(0)}]^\top (\widetilde{\boldsymbol{\theta}}^{(j)} - \widetilde{\boldsymbol{\theta}}^{(0)}) - \mathbf{v}) + m\lambda(\widetilde{\boldsymbol{\theta}}^{(j)} - \widetilde{\boldsymbol{\theta}}^{(0)}) \big].$$

Next lemma provides perturbation bounds for $\mathbf{J}^{(j)}, \mathbf{H}^{(j)}$ and $\|\mathbf{f}^{(j+1)} - \mathbf{f}^{(j)} - [\mathbf{J}^{(j)}]^\top (\boldsymbol{\theta}^{(j+1)} - \boldsymbol{\theta}^{(j)})\|_2$.

**Lemma C.2.** There exist constants $\{\bar{C}_i\}_{i=1}^6 > 0$ such that for any $\delta > 0$, if $\tau$ satisfies that

$$\bar{C}_1 m^{-3/2} L^{-3/2} [\log(TKL^2/\delta)]^{3/2} \le \tau \le \bar{C}_2 L^{-6} [\log m]^{-3/2},$$

then with probability at least $1 - \delta$ over the random initialization of $\boldsymbol{\theta}^{(0)}$, if for any $j \in [J]$, $\|\boldsymbol{\theta}^{(j)} - \boldsymbol{\theta}^{(0)}\|_2 \le \tau$, we have the following inequalities for any $j \in [J]$,

$$\big\| \mathbf{J}^{(j)} \big\|_F \le \bar{C}_4 \sqrt{tmL}, \tag{C.3}$$

$$\| \mathbf{J}^{(j)} - \mathbf{J}^{(0)} \|_F \le \bar{C}_5 \sqrt{tm \log m} \tau^{1/3} L^{7/2}, \tag{C.4}$$

$$\big\| \mathbf{f}^{(j+1)} - \mathbf{f}^{(j)} - [\mathbf{J}^{(j)}]^\top (\boldsymbol{\theta}^{(j+1)} - \boldsymbol{\theta}^{(j)}) \big\|_2 \le \bar{C}_6 \tau^{4/3} L^3 \sqrt{tm \log m}, \tag{C.5}$$

$$\| \mathbf{v} \|_2 \le \sqrt{t}. \tag{C.6}$$

Next lemma gives an upper bound for $\|\mathbf{f}^{(j)} - \mathbf{v}\|_2$.

**Lemma C.3.** There exist constants $\{\bar{C}_i\}_{i=1}^4 > 0$ such that for any $\delta > 0$, if $\tau, \eta$ satisfy that

$$\bar{C}_1 m^{-3/2} L^{-3/2} [\log(TKL^2/\delta)]^{3/2} \le \tau \le \bar{C}_2 L^{-6} [\log m]^{-3/2}, ,$$

$$\eta \le \bar{C}_3 (m\lambda + tmL)^{-1},$$

$$\tau^{8/3} \le \bar{C}_4 m(\lambda\eta)^2 L^{-6} t^{-1} (\log m)^{-1},$$

then with probability at least $1 - \delta$ over the random initialization of $\boldsymbol{\theta}^{(0)}$, if for any $j \in [J]$, $\|\boldsymbol{\theta}^{(j)} - \boldsymbol{\theta}^{(0)}\|_2 \le \tau$, we have that for any $j \in [J]$, $\|\mathbf{f}^{(j)} - \mathbf{v}\|_2 \le 2\sqrt{t}$.

Next lemma gives an upper bound of the distance between auxiliary sequence $\|\widetilde{\boldsymbol{\theta}}^{(j)} - \widetilde{\boldsymbol{\theta}}^{(0)}\|_2$.

**Lemma C.4.** There exist constants $\{\bar{C}_i\}_{i=1}^3 > 0$ such that for any $\delta \in (0,1)$, if $\tau, \eta$ satisfy that

$$\bar{C}_1 m^{-3/2} L^{-3/2} [\log(TKL^2/\delta)]^{3/2} \le \tau \le \bar{C}_2 L^{-6} [\log m]^{-3/2}, ,$$

$$\eta \le \bar{C}_3 (tmL + m\lambda)^{-1},$$

then with probability at least $1 - \delta$, we have that for any $j \in [J]$,

$$\big\| \widetilde{\boldsymbol{\theta}}^{(j)} - \boldsymbol{\theta}^{(0)} \big\|_2 \le \sqrt{t/(m\lambda)},$$

$$\big\| \widetilde{\boldsymbol{\theta}}^{(j)} - \boldsymbol{\theta}^{(0)} - (\bar{\mathbf{Z}})^{-1} \bar{\mathbf{b}}/\sqrt{m} \big\|_2 \le (1 - \eta m\lambda)^J \sqrt{t/(m\lambda)}$$

With above lemmas, we prove Lemma B.2 as follows.

*Proof of Lemma B.2.* Set $\tau = 2\sqrt{t/(m\lambda)}$. First we assume that $\|\boldsymbol{\theta}^{(j)} - \boldsymbol{\theta}^{(0)}\|_2 \leq \tau$ for all $0 \leq j \leq J$. Then with this assumption and the choice of $m, \tau$, we have that Lemma C.2, C.3 and C.4 hold. Then we have

$$
\begin{aligned}
\big\|\boldsymbol{\theta}^{(j+1)} - \widetilde{\boldsymbol{\theta}}^{(j+1)}\big\|_2 &= \big\|\boldsymbol{\theta}^{(j)} - \widetilde{\boldsymbol{\theta}}^{(j)} - \eta(\mathbf{J}^{(j)} - \mathbf{J}^{(0)})(\mathbf{f}^{(j)} - \mathbf{v}) - \eta m\lambda(\boldsymbol{\theta}^{(j)} - \widetilde{\boldsymbol{\theta}}^{(j)}) \\
&\quad - \eta\mathbf{J}^{(0)}(\mathbf{f}^{(j)} - [\mathbf{J}^{(0)}]^\top(\widetilde{\boldsymbol{\theta}}^{(j)} - \boldsymbol{\theta}^{(0)}))\big\|_2 \\
&= \big\|(1 - \eta m\lambda)(\boldsymbol{\theta}^{(j)} - \widetilde{\boldsymbol{\theta}}^{(j)}) - \eta(\mathbf{J}^{(j)} - \mathbf{J}^{(0)})(\mathbf{f}^{(j)} - \mathbf{v}) \\
&\quad - \eta\mathbf{J}^{(0)}\big[\mathbf{f}^{(j)} - [\mathbf{J}^{(0)}](\boldsymbol{\theta}^{(j)} - \boldsymbol{\theta}^{(0)}) + [\mathbf{J}^{(0)}]^\top(\boldsymbol{\theta}^{(j)} - \widetilde{\boldsymbol{\theta}}^{(j)})\big]\big\|_2 \\
&\leq \big\|\big[\mathbf{I} - \eta(m\lambda\mathbf{I} + \mathbf{H}^{(0)})\big](\widetilde{\boldsymbol{\theta}}^{(j)} - \boldsymbol{\theta}^{(j)})\big\|_2 + \eta\big\|(\mathbf{J}^{(j)} - \mathbf{J}^{(0)})(\mathbf{f}^{(j)} - \mathbf{v})\big\|_2 \\
&\quad + \eta\|\mathbf{J}^{(0)}\|_2\big\|\mathbf{f}^{(j)} - [\mathbf{J}^{(0)}](\boldsymbol{\theta}^{(j)} - \boldsymbol{\theta}^{(0)})\big\|_2 \\
&\leq (1 - \eta m\lambda)\big\|\widetilde{\boldsymbol{\theta}}^{(j)} - \boldsymbol{\theta}^{(j)}\big\|_2 + \underbrace{\eta\big\|(\mathbf{J}^{(j)} - \mathbf{J}^{(0)})(\mathbf{f}^{(j)} - \mathbf{v})\big\|_2}_{I_1} \\
&\quad + \underbrace{\eta\|\mathbf{J}^{(0)}\|_2\big\|\mathbf{f}^{(j)} - [\mathbf{J}^{(0)}](\boldsymbol{\theta}^{(j)} - \boldsymbol{\theta}^{(0)})\big\|_2}_{I_2},
\end{aligned}
\tag{C.7}
$$

where the first inequality holds due to triangle inequality, the second inequality holds due to the fact that $\eta\mathbf{H}^{(0)} = \eta[\mathbf{J}^{(0)}]^\top\mathbf{J}^{(0)} \preceq C_1 tmL\eta\mathbf{I} \leq \mathbf{I}$ with (C.3) in Lemma C.2 for some $C_1 > 0$. We now bound $I_1, I_2$ separately. For $I_1$, we have

$$
I_1 \leq \eta\big\|\mathbf{J}^{(j)} - \mathbf{J}^{(0)}\big\|_2\|\mathbf{f}^{(j)} - \mathbf{v}\|_2 \leq \eta C_2 t\sqrt{m\log m}\tau^{1/3}L^{7/2},
\tag{C.8}
$$

where $C_2 > 0$ is a constant, the first inequality holds due to matrix spectral norm and the second inequality holds due to (C.4) in Lemma C.2 and Lemma C.3. For $I_2$, we have

$$
I_2 \leq \eta\big\|\mathbf{J}^{(0)}\big\|_2\big\|\mathbf{f}^{(j)} - \mathbf{J}^{(0)}(\boldsymbol{\theta}^{(j)} - \boldsymbol{\theta}^{(0)})\big\|_2 \leq \eta C_3 tmL^{7/2}\tau^{4/3}\sqrt{\log m},
\tag{C.9}
$$

where $C_3 > 0$, the first inequality holds due to matrix spectral norm, the second inequality holds due to (C.3) and (C.5) in Lemma C.2 and the fact that $\mathbf{f}^{(0)} = \mathbf{0}$ by random initialization over $\boldsymbol{\theta}^{(0)}$. Substituting (C.8) and (C.9) into (C.7), we have

$$
\begin{aligned}
\big\|\boldsymbol{\theta}^{(j+1)} &- \widetilde{\boldsymbol{\theta}}^{(j+1)}\big\|_2 \\
&\leq (1 - \eta m\lambda)\big\|\boldsymbol{\theta}^{(j)} - \widetilde{\boldsymbol{\theta}}^{(j)}\big\|_2 + C_4\big(\eta t\sqrt{m\log m}\tau^{1/3}L^{7/2} + \eta tmL^{7/2}\tau^{4/3}\sqrt{\log m}\big),
\end{aligned}
\tag{C.10}
$$

where $C_4 > 0$ is a constant. Expanding (C.10) for $k$ times, we have

$$
\begin{aligned}
\big\|\boldsymbol{\theta}^{(j+1)} - \widetilde{\boldsymbol{\theta}}^{(j+1)}\big\|_2 &\leq C_4 \frac{\eta t\sqrt{m\log m}\tau^{1/3}L^{7/2} + \eta tmL^{7/2}\tau^{4/3}\sqrt{\log m}}{\eta m\lambda} \\
&= C_5 m^{-2/3}\sqrt{\log m}L^{7/2}t^{5/3}\lambda^{-5/3}(1 + \sqrt{t/\lambda}) \\
&\leq \frac{\tau}{2},
\end{aligned}
\tag{C.11}
$$

where $C_5 > 0$ is a constant, the equality holds by the definition of $\tau$, the last inequality holds due to the choice of $m$, where

$$
m^{1/6} \geq C_6\sqrt{\log m}L^{7/2}t^{7/6}\lambda^{-7/6}(1 + \sqrt{t/\lambda}),
$$

$C_6 > 0$ is a constant. Thus, for any $j \in [J]$, we have

$$
\|\boldsymbol{\theta}^{(j)} - \boldsymbol{\theta}^{(0)}\|_2 \leq \|\widetilde{\boldsymbol{\theta}}^{(j)} - \boldsymbol{\theta}^{(0)}\|_2 + \|\boldsymbol{\theta}^{(j)} - \widetilde{\boldsymbol{\theta}}^{(j)}\|_2 \leq \sqrt{t/(m\lambda)} + \tau/2 = \tau,
\tag{C.12}
$$

where the first inequality holds due to triangle inequality, the second inequality holds due to Lemma C.4. (C.13) suggests that our assumption $\|\boldsymbol{\theta}^{(j)} - \boldsymbol{\theta}^{(0)}\|_2 \leq \tau$ holds for any $u$. Note that we have the following inequality by Lemma C.4:

$$
\big\|\widetilde{\boldsymbol{\theta}}^{(j)} - \boldsymbol{\theta}^{(0)} - (\bar{\mathbf{Z}})^{-1}\bar{\mathbf{b}}/\sqrt{m}\big\|_2 \leq (1 - \eta m\lambda)^j\sqrt{t/(m\lambda)}.
\tag{C.13}
$$

Using (C.11) and (C.13), we have

$$
\begin{aligned}
\big\|\boldsymbol{\theta}^{(j)} &- \boldsymbol{\theta}^{(0)} - (\bar{\mathbf{Z}})^{-1}\bar{\mathbf{b}}/\sqrt{m}\big\|_2 \\
&\leq (1 - \eta m\lambda)^J\sqrt{t/(m\lambda)} + C_5 m^{-2/3}\sqrt{\log m}L^{7/2}t^{5/3}\lambda^{-5/3}(1 + \sqrt{t/\lambda}).
\end{aligned}
$$

This completes the proof. $\qquad\square$

### C.3 PROOF OF LEMMA B.3

In this section we prove Lemma B.3.

*Proof of Lemma B.3.* Set $\tau = 2\sqrt{t/(m\lambda)}$. By Lemma 6.2, we have that $\|\boldsymbol{\theta}_i - \boldsymbol{\theta}_0\|_2 \le \tau$ for $i \in [t]$. $\|\mathbf{Z}_t\|_F$ can be bounded as follows.

$$\|\mathbf{Z}_t\|_F = \left\| \sum_{i=1}^{t} \mathbf{g}(\mathbf{x}_{i,a_i};\boldsymbol{\theta}_i)\mathbf{g}(\mathbf{x}_{i,a_i};\boldsymbol{\theta}_i)^\top/m \right\|_F \le \sum_{i=1}^{t} \left\|\mathbf{g}(\mathbf{x}_{i,a_i};\boldsymbol{\theta}_i)\right\|_2^2/m \le C_0 t L,$$

where $C_0 > 0$ is a constant, the first inequality holds due to the fact that $\|\mathbf{a}\mathbf{a}^\top\|_F = \|\mathbf{a}\|_2^2$, the second inequality holds due to Lemma B.6 with the fact that $\|\boldsymbol{\theta}_i - \boldsymbol{\theta}_0\|_2 \le \tau$. We bound $\|\mathbf{Z}_t - \bar{\mathbf{Z}}_t\|_2$ as follows. We have

$$\begin{aligned}
\|\mathbf{Z}_t - \bar{\mathbf{Z}}_t\|_F &= \left\| \sum_{i=1}^{t} \Big( \mathbf{g}(\mathbf{x}_{i,a_i};\boldsymbol{\theta}_0)\mathbf{g}(\mathbf{x}_{i,a_i};\boldsymbol{\theta}_0)^\top - \mathbf{g}(\mathbf{x}_{i,a_i};\boldsymbol{\theta}_i)\mathbf{g}(\mathbf{x}_{i,a_i};\boldsymbol{\theta}_i)^\top \Big)/m \right\|_F \\
&\le \sum_{i=1}^{t} \left\| \mathbf{g}(\mathbf{x}_{i,a_i};\boldsymbol{\theta}_0)\mathbf{g}(\mathbf{x}_{i,a_i};\boldsymbol{\theta}_0)^\top - \mathbf{g}(\mathbf{x}_{i,a_i};\boldsymbol{\theta}_i)\mathbf{g}(\mathbf{x}_{i,a_i};\boldsymbol{\theta}_i)^\top \right\|_F/m \\
&\le \sum_{i=1}^{t} \Big( \left\|\mathbf{g}(\mathbf{x}_{i,a_i};\boldsymbol{\theta}_0)\right\|_2 + \left\|\mathbf{g}(\mathbf{x}_{i,a_i};\boldsymbol{\theta}_i)\right\|_2 \Big) \left\|\mathbf{g}(\mathbf{x}_{i,a_i};\boldsymbol{\theta}_0) - \mathbf{g}(\mathbf{x}_{i,a_i};\boldsymbol{\theta}_i)\right\|_2/m,
\end{aligned} \tag{C.14}$$

where the first inequality holds due to triangle inequality, the second inequality holds the fact that $\|\mathbf{a}\mathbf{a}^\top - \mathbf{b}\mathbf{b}^\top\|_F \le (\|\mathbf{a}\|_2 + \|\mathbf{b}\|_2)\|\mathbf{a} - \mathbf{b}\|_2$ for any vectors $\mathbf{a}, \mathbf{b}$. To bound (C.14), we have

$$\left\|\mathbf{g}(\mathbf{x}_{i,a_i};\boldsymbol{\theta}_0)\right\|_2, \left\|\mathbf{g}(\mathbf{x}_{i,a_i};\boldsymbol{\theta}_i)\right\|_2 \le C_1\sqrt{mL}, \tag{C.15}$$

where $C_1 > 0$ is a constant, the inequality holds due to Lemma B.6 with the fact that $\|\boldsymbol{\theta}_i - \boldsymbol{\theta}_0\|_2 \le \tau$. We also have

$$\left\|\mathbf{g}(\mathbf{x}_{i,a_i};\boldsymbol{\theta}_0) - \mathbf{g}(\mathbf{x}_{i,a_i};\boldsymbol{\theta}_i)\right\|_2 \le C_2\sqrt{\log m}\tau^{1/3}L^3\|\mathbf{g}(\mathbf{x}_j;\boldsymbol{\theta}_0)\|_2 \le C_3\sqrt{m\log m}\tau^{1/3}L^{7/2}, \tag{C.16}$$

where $C_2, C_3 > 0$ are constants, the first inequality holds due to Lemma B.5 with the fact that $\|\boldsymbol{\theta}_i - \boldsymbol{\theta}_0\|_2 \le \tau$, the second inequality holds due to Lemma B.6. Substituting (C.15) and (C.16) into (C.14), we have

$$\|\mathbf{Z}_t - \bar{\mathbf{Z}}_t\|_F \le C_4 t\sqrt{\log m}\tau^{1/3}L^4,$$

where $C_4 > 0$ is a constant. We now bound $\log\det\bar{\mathbf{Z}}_t - \log\det\mathbf{Z}_t$. It is easy to verify that $\bar{\mathbf{Z}}_t = \lambda\mathbf{I} + \bar{\mathbf{J}}[\bar{\mathbf{J}}]^\top$, $\mathbf{Z}_t = \lambda\mathbf{I} + \mathbf{J}\mathbf{J}^\top$, where

$$\bar{\mathbf{J}} = \Big( \mathbf{g}(\mathbf{x}_{1,a_1};\boldsymbol{\theta}_0), \ldots, \mathbf{g}(\mathbf{x}_{t,a_t};\boldsymbol{\theta}_0) \Big)/\sqrt{m},$$

$$\mathbf{J} = \Big( \mathbf{g}(\mathbf{x}_{1,a_1};\boldsymbol{\theta}_0), \ldots, \mathbf{g}(\mathbf{x}_{t,a_t};\boldsymbol{\theta}_{t-1}) \Big)/\sqrt{m}.$$

We have the following inequalities:

$$\begin{aligned}
\log\frac{\det(\bar{\mathbf{Z}}_t)}{\det(\lambda\mathbf{I})} - \log\frac{\det(\mathbf{Z}_t)}{\det(\lambda\mathbf{I})} &= \log\det(\mathbf{I} + \bar{\mathbf{J}}[\bar{\mathbf{J}}]^\top/\lambda) - \log\det(\mathbf{I} + \mathbf{J}[\mathbf{J}]^\top/\lambda) \\
&= \log\det(\mathbf{I} + [\bar{\mathbf{J}}]^\top\bar{\mathbf{J}}/\lambda) - \log\det(\mathbf{I} + [\mathbf{J}]^\top\mathbf{J}/\lambda) \\
&\le \langle (\mathbf{I} + [\mathbf{J}]^\top\mathbf{J}/\lambda)^{-1}, [\bar{\mathbf{J}}]^\top\bar{\mathbf{J}} - [\mathbf{J}]^\top\mathbf{J} \rangle \\
&\le \|(\mathbf{I} + [\mathbf{J}]^\top\mathbf{J}/\lambda)^{-1}\|_F \|[\bar{\mathbf{J}}]^\top\bar{\mathbf{J}} - [\mathbf{J}]^\top\mathbf{J}\|_F \\
&\le \sqrt{t}\|(\mathbf{I} + [\mathbf{J}]^\top\mathbf{J}/\lambda)^{-1}\|_2 \|[\bar{\mathbf{J}}]^\top\bar{\mathbf{J}} - [\mathbf{J}]^\top\mathbf{J}\|_F \\
&\le \sqrt{t}\|[\bar{\mathbf{J}}]^\top\bar{\mathbf{J}} - [\mathbf{J}]^\top\mathbf{J}\|_F, \tag{C.17}
\end{aligned}$$

where the second equality holds due to the fact that $\det(\mathbf{I} + \mathbf{A}\mathbf{A}^\top) = \det(\mathbf{I} + \mathbf{A}^\top\mathbf{A})$, the first inequality holds due to the fact that $\log \det$ function is convex, the second inequality hold due to the fact that $\langle \mathbf{A}, \mathbf{B} \rangle \leq \|\mathbf{A}\|_F \|\mathbf{B}\|_F$, the third inequality holds since $\mathbf{I} + [\mathbf{J}]^\top \mathbf{J}/\lambda$ is a $t$-dimension matrix, the fourth inequality holds since $\mathbf{I} + [\mathbf{J}]^\top \mathbf{J}/\lambda \succeq \mathbf{I}$. We have

$$
\begin{aligned}
&\|[\bar{\mathbf{J}}]^\top \bar{\mathbf{J}} - [\mathbf{J}]^\top \mathbf{J}\|_F \\
&\leq t \max_{1 \leq i,j \leq t} \left| \mathbf{g}(\mathbf{x}_{i,a_i}; \boldsymbol{\theta}_0)^\top \mathbf{g}(\mathbf{x}_{j,a_j}; \boldsymbol{\theta}_0) - \mathbf{g}(\mathbf{x}_{i,a_i}; \boldsymbol{\theta}_i)^\top \mathbf{g}(\mathbf{x}_{j,a_j}; \boldsymbol{\theta}_j) \right|/m \\
&\leq t \max_{1 \leq i,j \leq t} \left\| \mathbf{g}(\mathbf{x}_{i,a_i}; \boldsymbol{\theta}_0) - \mathbf{g}(\mathbf{x}_{i,a_i}; \boldsymbol{\theta}_i) \right\|_2 \left\| \mathbf{g}(\mathbf{x}_{j,a_j}; \boldsymbol{\theta}_j) \right\|_2/m \\
&\quad + \left\| \mathbf{g}(\mathbf{x}_{j,a_j}; \boldsymbol{\theta}_0) - \mathbf{g}(\mathbf{x}_{j,a_j}; \boldsymbol{\theta}_j) \right\|_2 \left\| \mathbf{g}(\mathbf{x}_{i,a_i}; \boldsymbol{\theta}_0) \right\|_2/m \\
&\leq C_5 t \sqrt{\log m} \tau^{1/3} L^4,
\end{aligned}
\tag{C.18}
$$

where $C_5 > 0$ is a constant, the first inequality holds due to the fact that $\|\mathbf{A}\|_F \leq t \max |\mathbf{A}_{i,j}|$ for any $\mathbf{A} \in \mathbb{R}^{t \times t}$, the second inequality holds due to the fact $|\mathbf{a}^\top \mathbf{a}' - \mathbf{b}^\top \mathbf{b}'| \leq \|\mathbf{a} - \mathbf{b}\|_2 \|\mathbf{b}'\|_2 + \|\mathbf{a}' - \mathbf{b}'\|_2 \|\mathbf{a}\|_2$, the third inequality holds due to (C.15) and (C.16). Substituting (C.18) into (C.17), we have

$$
\log \frac{\det(\bar{\mathbf{Z}}_t)}{\det(\lambda \mathbf{I})} - \log \frac{\det(\mathbf{Z}_t)}{\det(\lambda \mathbf{I})} \leq C_5 t^{3/2} \sqrt{\log m} \tau^{1/3} L^4.
$$

Using the same method, we also have

$$
\log \frac{\det(\mathbf{Z}_t)}{\det(\lambda \mathbf{I})} - \log \frac{\det(\bar{\mathbf{Z}}_t)}{\det(\lambda \mathbf{I})} \leq C_5 t^{3/2} \sqrt{\log m} \tau^{1/3} L^4.
$$

This completes our proof.

$\square$

# D   PROOFS OF LEMMAS IN APPENDIX C

## D.1   PROOF OF LEMMA C.2

In this section we give the proof of Lemma C.2.

*Proof of Lemma C.2.* For any $j \in [J]$, the following inequalities hold. We first have

$$
\left\| \mathbf{J}^{(j)} \right\|_F \leq \sqrt{t} \max_{i \in [t]} \left\| \mathbf{g}(\mathbf{x}_{i,a_i}; \boldsymbol{\theta}^{(j)}) \right\|_2 \leq C_1 \sqrt{tmL},
\tag{D.1}
$$

where $C_1 > 0$ is a constant, the first inequality holds due to the definition of $\mathbf{J}^{(j)}$, the second inequality holds due to Lemma B.6. We also have

$$
\|\mathbf{J}^{(j)} - \mathbf{J}^{(0)}\|_F \leq C_2 \sqrt{\log m} \tau^{1/3} L^3 \|\mathbf{J}^{(0)}\|_F \leq C_3 \sqrt{tm \log m} \tau^{1/3} L^{7/2},
\tag{D.2}
$$

where $C_2, C_3 > 0$ are constants, the first inequality holds due to Lemma B.5 with the assumption that $\|\boldsymbol{\theta}^{(j)} - \boldsymbol{\theta}^{(0)}\|_2 \leq \tau$, the second inequality holds due to (D.1). We also have

$$
\begin{aligned}
&\left\| \mathbf{f}^{(j+1)} - \mathbf{f}^{(j)} - [\mathbf{J}^{(j)}]^\top (\boldsymbol{\theta}^{(j+1)} - \boldsymbol{\theta}^{(j)}) \right\|_2 \\
&\leq \max_{i \in [t]} \sqrt{t} \left| f(\mathbf{x}_{i,a_i}; \boldsymbol{\theta}^{(j+1)}) - f(\mathbf{x}_{i,a_i}; \boldsymbol{\theta}^{(j)}) - \langle \mathbf{g}(\mathbf{x}_{i,a_i}; \boldsymbol{\theta}^{(j)}), \boldsymbol{\theta}^{(j+1)} - \boldsymbol{\theta}^{(j)} \rangle \right| \\
&\leq C_4 \tau^{4/3} L^3 \sqrt{tm \log m},
\end{aligned}
$$

where $C_4 > 0$ is a constant, the first inequality holds due to the the fact that $\|\mathbf{x}\|_2 \leq \sqrt{t} \max |x_i|$ for any $\mathbf{x} \in \mathbb{R}^t$, the second inequality holds due to Lemma B.4 with the assumption that $\|\boldsymbol{\theta}^{(j)} - \boldsymbol{\theta}^{(0)}\|_2 \leq \tau$, $\|\boldsymbol{\theta}^{(j+1)} - \boldsymbol{\theta}^{(0)}\|_2 \leq \tau$. For $\|\mathbf{v}\|_2$, we have $\|\mathbf{v}\|_2 \leq \sqrt{t} \max_{1 \leq i \leq t} |r(\mathbf{x}_{i,a_i})| \leq \sqrt{t}$. This completes our proof.

$\square$

## D.2 PROOF OF LEMMA C.3

*Proof of Lemma C.3.* Recall that the loss function $L$ is defined as

$$L(\boldsymbol{\theta}) = \frac{1}{2}\|\mathbf{f}(\boldsymbol{\theta}) - \mathbf{v}\|_2^2 + \frac{m\lambda}{2}\|\boldsymbol{\theta} - \boldsymbol{\theta}^{(0)}\|_2^2.$$

We define $\mathbf{J}(\boldsymbol{\theta})$ and $\mathbf{f}(\boldsymbol{\theta})$ as follows:

$$\mathbf{J}(\boldsymbol{\theta}) = \Big(\mathbf{g}(\mathbf{x}_{1,a_1};\boldsymbol{\theta}),\ldots,\mathbf{g}(\mathbf{x}_{t,a_t};\boldsymbol{\theta})\Big) \in \mathbb{R}^{(md+m^2(L-2)+m)\times t},$$

$$\mathbf{f}(\boldsymbol{\theta}) = (f(\mathbf{x}_{1,a_1};\boldsymbol{\theta}),\ldots,f(\mathbf{x}_{t,a_t};\boldsymbol{\theta}))^\top \in \mathbb{R}^{t\times 1}.$$

Suppose $\|\boldsymbol{\theta} - \boldsymbol{\theta}^{(0)}\|_2 \leq \tau$. Then by the fact that $\|\cdot\|_2^2/2$ is 1-strongly convex and 1-smooth, we have the following inequalities:

$$L(\boldsymbol{\theta}') - L(\boldsymbol{\theta})$$

$$\leq \langle \mathbf{f}(\boldsymbol{\theta}) - \mathbf{v}, \mathbf{f}(\boldsymbol{\theta}') - \mathbf{f}(\boldsymbol{\theta})\rangle + \frac{1}{2}\|\mathbf{f}(\boldsymbol{\theta}') - \mathbf{f}(\boldsymbol{\theta})\|_2^2 + m\lambda\langle\boldsymbol{\theta} - \boldsymbol{\theta}^{(0)}, \boldsymbol{\theta}' - \boldsymbol{\theta}\rangle + \frac{m\lambda}{2}\|\boldsymbol{\theta}' - \boldsymbol{\theta}\|_2^2$$

$$= \langle \mathbf{f}(\boldsymbol{\theta}) - \mathbf{v}, [\mathbf{J}(\boldsymbol{\theta})]^\top(\boldsymbol{\theta}' - \boldsymbol{\theta}) + \mathbf{e}\rangle + \frac{1}{2}\|[\mathbf{J}(\boldsymbol{\theta})]^\top(\boldsymbol{\theta}' - \boldsymbol{\theta}) + \mathbf{e}\|_2^2$$

$$+ m\lambda\langle\boldsymbol{\theta} - \boldsymbol{\theta}^{(0)}, \boldsymbol{\theta}' - \boldsymbol{\theta}\rangle + \frac{m\lambda}{2}\|\boldsymbol{\theta}' - \boldsymbol{\theta}\|_2^2$$

$$= \langle \mathbf{J}(\boldsymbol{\theta})(\mathbf{f}(\boldsymbol{\theta}) - \mathbf{v}) + m\lambda(\boldsymbol{\theta} - \boldsymbol{\theta}^{(0)}), \boldsymbol{\theta}' - \boldsymbol{\theta}\rangle + \langle\mathbf{f}(\boldsymbol{\theta}) - \mathbf{v}, \mathbf{e}\rangle$$

$$+ \frac{1}{2}\|[\mathbf{J}(\boldsymbol{\theta})]^\top(\boldsymbol{\theta}' - \boldsymbol{\theta}) + \mathbf{e}\|_2^2 + \frac{m\lambda}{2}\|\boldsymbol{\theta}' - \boldsymbol{\theta}\|_2^2$$

$$= \langle\nabla L(\boldsymbol{\theta}), \boldsymbol{\theta}' - \boldsymbol{\theta}\rangle + \underbrace{\langle\mathbf{f}(\boldsymbol{\theta}) - \mathbf{v}, \mathbf{e}\rangle + \frac{1}{2}\|[\mathbf{J}(\boldsymbol{\theta})]^\top(\boldsymbol{\theta}' - \boldsymbol{\theta}) + \mathbf{e}\|_2^2 + \frac{m\lambda}{2}\|\boldsymbol{\theta}' - \boldsymbol{\theta}\|_2^2}_{I_1}, \qquad \text{(D.3)}$$

where $\mathbf{e} = \mathbf{f}(\boldsymbol{\theta}') - \mathbf{f}(\boldsymbol{\theta}) - \mathbf{J}(\boldsymbol{\theta})^\top(\boldsymbol{\theta}' - \boldsymbol{\theta})$. $I_1$ can be bounded as follows:

$$I_1 \leq \|\mathbf{f}(\boldsymbol{\theta}) - \mathbf{v}\|_2\|\mathbf{e}\|_2 + \|\mathbf{J}(\boldsymbol{\theta})\|_2^2\|\boldsymbol{\theta}' - \boldsymbol{\theta}\|_2^2 + \|\mathbf{e}\|_2^2 + \frac{m\lambda}{2}\|\boldsymbol{\theta}' - \boldsymbol{\theta}\|_2^2$$

$$\leq \frac{C_1}{2}\Big((m\lambda + tmL)\|\boldsymbol{\theta}' - \boldsymbol{\theta}\|_2^2\Big) + \|\mathbf{f}(\boldsymbol{\theta}) - \mathbf{v}\|_2\|\mathbf{e}\|_2 + \|\mathbf{e}\|_2^2, \qquad \text{(D.4)}$$

where the first inequality holds due to Cauchy-Schwarz inequality, the second inequality holds due to the fact that $\|\mathbf{J}(\boldsymbol{\theta})\|_2 \leq C_2\sqrt{tmL}$ with $\|\boldsymbol{\theta} - \boldsymbol{\theta}^{(0)}\|_2 \leq \tau$ by (C.3) in Lemma C.2. Substituting (D.4) into (D.3), we have

$$L(\boldsymbol{\theta}') - L(\boldsymbol{\theta}) \leq \langle\nabla L(\boldsymbol{\theta}), \boldsymbol{\theta}' - \boldsymbol{\theta}\rangle + \frac{C_1}{2}\Big((m\lambda + tmL)\|\boldsymbol{\theta}' - \boldsymbol{\theta}\|_2^2\Big) + \|\mathbf{f}(\boldsymbol{\theta}) - \mathbf{v}\|_2\|\mathbf{e}\|_2 + \|\mathbf{e}\|_2^2.$$

$$\text{(D.5)}$$

Taking $\boldsymbol{\theta}' = \boldsymbol{\theta} - \eta\nabla L(\boldsymbol{\theta})$, then by (D.5), we have

$$L(\boldsymbol{\theta} - \eta\nabla L(\boldsymbol{\theta})) - L(\boldsymbol{\theta}) \leq -\eta\|\nabla L(\boldsymbol{\theta})\|_2^2(1 - C_1(m\lambda + tmL)\eta) + \|\mathbf{f}(\boldsymbol{\theta}) - \mathbf{v}\|_2\|\mathbf{e}\|_2 + \|\mathbf{e}\|_2^2.$$

$$\text{(D.6)}$$

By the 1-strongly convexity of $\|\cdot\|_2^2$, we further have

$$L(\boldsymbol{\theta}') - L(\boldsymbol{\theta})$$

$$\geq \langle\mathbf{f}(\boldsymbol{\theta}) - \mathbf{v}, \mathbf{f}(\boldsymbol{\theta}') - \mathbf{f}(\boldsymbol{\theta})\rangle + m\lambda\langle\boldsymbol{\theta} - \boldsymbol{\theta}^{(0)}, \boldsymbol{\theta}' - \boldsymbol{\theta}\rangle + \frac{m\lambda}{2}\|\boldsymbol{\theta}' - \boldsymbol{\theta}\|_2^2$$

$$= \langle\mathbf{f}(\boldsymbol{\theta}) - \mathbf{v}, [\mathbf{J}(\boldsymbol{\theta})]^\top(\boldsymbol{\theta}' - \boldsymbol{\theta}) + \mathbf{e}\rangle + m\lambda\langle\boldsymbol{\theta} - \boldsymbol{\theta}^{(0)}, \boldsymbol{\theta}' - \boldsymbol{\theta}\rangle + \frac{m\lambda}{2}\|\boldsymbol{\theta}' - \boldsymbol{\theta}\|_2^2$$

$$= \langle\nabla L(\boldsymbol{\theta}), \boldsymbol{\theta}' - \boldsymbol{\theta}\rangle + \frac{m\lambda}{2}\|\boldsymbol{\theta}' - \boldsymbol{\theta}\|_2^2 + \langle\mathbf{f}(\boldsymbol{\theta}) - \mathbf{v}, \mathbf{e}\rangle$$

$$\geq \langle\nabla L(\boldsymbol{\theta}), \boldsymbol{\theta}' - \boldsymbol{\theta}\rangle + \frac{m\lambda}{2}\|\boldsymbol{\theta}' - \boldsymbol{\theta}\|_2^2 - \|\mathbf{f}(\boldsymbol{\theta}) - \mathbf{v}\|_2\|\mathbf{e}\|_2$$

$$\geq -\frac{\|\nabla L(\boldsymbol{\theta})\|_2^2}{2m\lambda} - \|\mathbf{f}(\boldsymbol{\theta}) - \mathbf{v}\|_2\|\mathbf{e}\|_2, \qquad \text{(D.7)}$$

where the second inequality holds due to Cauchy-Schwarz inequality, the last inequality holds due to the fact that $\langle \mathbf{a}, \mathbf{x} \rangle + c\|\mathbf{x}\|_2^2 \geq -\|\mathbf{a}\|_2^2/(4c)$ for any vectors $\mathbf{a}, \mathbf{x}$ and $c > 0$. Substituting (D.7) into (D.6), we have

$$L(\boldsymbol{\theta} - \eta \nabla L(\boldsymbol{\theta})) - L(\boldsymbol{\theta})$$

$$\leq 2m\lambda\eta(1 - C_1(m\lambda + tmL)\eta)\big[L(\boldsymbol{\theta}') - L(\boldsymbol{\theta}) + \|\mathbf{f}(\boldsymbol{\theta}) - \mathbf{v}\|_2\|\mathbf{e}\|_2\big] + \|\mathbf{f}(\boldsymbol{\theta}) - \mathbf{v}\|_2\|\mathbf{e}\|_2 + \|\mathbf{e}\|_2^2$$

$$\leq m\lambda\eta\big[L(\boldsymbol{\theta}') - L(\boldsymbol{\theta}) + \|\mathbf{f}(\boldsymbol{\theta}) - \mathbf{v}\|_2\|\mathbf{e}\|_2\big] + \|\mathbf{f}(\boldsymbol{\theta}) - \mathbf{v}\|_2\|\mathbf{e}\|_2 + \|\mathbf{e}\|_2^2$$

$$\leq m\lambda\eta\big[L(\boldsymbol{\theta}') - L(\boldsymbol{\theta}) + \|\mathbf{f}(\boldsymbol{\theta}) - \mathbf{v}\|_2^2/8 + 4\|\mathbf{e}\|_2^2\big] + m\lambda\eta\|\mathbf{f}(\boldsymbol{\theta}) - \mathbf{v}\|_2^2/8 + 4\|\mathbf{e}\|_2^2/(m\lambda\eta) + \|\mathbf{e}\|_2^2$$

$$\leq m\lambda\eta(L(\boldsymbol{\theta}') - L(\boldsymbol{\theta})/2) + \|\mathbf{e}\|_2^2(1 + 4m\lambda\eta + 4/(m\lambda\eta)), \tag{D.8}$$

where the second inequality holds due to the choice of $\eta$, third inequality holds due to Young's inequality, fourth inequality holds due to the fact that $\|\mathbf{f}(\boldsymbol{\theta}) - \mathbf{v}\|_2^2 \leq 2L(\boldsymbol{\theta})$. Now taking $\boldsymbol{\theta} = \boldsymbol{\theta}^{(j)}$ and $\boldsymbol{\theta}' = \boldsymbol{\theta}^{(0)}$, rearranging (D.8), with the fact that $\boldsymbol{\theta}^{(j+1)} = \boldsymbol{\theta}^{(j)} - \eta \nabla L(\boldsymbol{\theta}^{(j)})$, we have

$$L(\boldsymbol{\theta}^{(j+1)}) - L(\boldsymbol{\theta}^{(0)})$$

$$\leq (1 - m\lambda\eta/2)[L(\boldsymbol{\theta}^{(j)}) - L(\boldsymbol{\theta}^{(0)})] + m\lambda\eta/2L(\boldsymbol{\theta}^{(0)}) + \|\mathbf{e}\|_2^2(1 + 4m\lambda\eta + 4/(m\lambda\eta))$$

$$\leq (1 - m\lambda\eta/2)[L(\boldsymbol{\theta}^{(j)}) - L(\boldsymbol{\theta}^{(0)})] + m\lambda\eta/2 \cdot t + m\lambda\eta/2 \cdot t$$

$$\leq (1 - m\lambda\eta/2)[L(\boldsymbol{\theta}^{(j)}) - L(\boldsymbol{\theta}^{(0)})] + m\lambda\eta t, \tag{D.9}$$

where the second inequality holds due to the fact that $L(\boldsymbol{\theta}^{(0)}) = \|\mathbf{f}(\boldsymbol{\theta}^{(0)}) - \mathbf{v}\|_2^2/2 = \|\mathbf{v}\|_2^2/2 \leq t$, and

$$(1 + 4m\lambda\eta + 4/(m\lambda\eta))\|\mathbf{e}\|_2^2 \leq 5/(m\lambda\eta) \cdot C_2\tau^{8/3}L^6 tm \log m \leq tm\lambda\eta/2, \tag{D.10}$$

where the first inequality holds due to (C.5) in Lemma C.2, the second inequality holds due to the choice of $\tau$. Expanding (D.9) $u$ times, we have

$$L(\boldsymbol{\theta}^{(j+1)}) - L(\boldsymbol{\theta}^{(0)}) \leq \frac{m\lambda\eta t}{m\lambda\eta/2} = 2t,$$

which implies that $\|\mathbf{f}^{(j+1)} - \mathbf{v}\|_2 \leq 2\sqrt{t}$. This completes our proof. $\qquad \square$

### D.3 Proof of Lemma C.4

In this section we prove Lemma C.4.

*Proof of Lemma C.4.* It is worth noting that $\widetilde{\boldsymbol{\theta}}^{(j)}$ is the sequence generated by applying gradient descent on the following problem:

$$\min_{\boldsymbol{\theta}} \widetilde{L}(\boldsymbol{\theta}) = \frac{1}{2}\|[\mathbf{J}^{(0)}]^\top(\boldsymbol{\theta} - \boldsymbol{\theta}^{(0)}) - \mathbf{v}\|_2^2 + \frac{m\lambda}{2}\|\boldsymbol{\theta} - \boldsymbol{\theta}^{(0)}\|_2^2.$$

Then $\|\boldsymbol{\theta}^{(0)} - \widetilde{\boldsymbol{\theta}}^{(j)}\|_2$ can be bounded as

$$\frac{m\lambda}{2}\|\boldsymbol{\theta}^{(0)} - \widetilde{\boldsymbol{\theta}}^{(j)}\|_2^2 \leq \frac{1}{2}\|[\mathbf{J}^{(0)}]^\top(\widetilde{\boldsymbol{\theta}}^{(j)} - \boldsymbol{\theta}^{(0)}) - \mathbf{v}\|_2^2 + \frac{m\lambda}{2}\|\widetilde{\boldsymbol{\theta}}^{(j)} - \boldsymbol{\theta}^{(0)}\|_2^2$$

$$\leq \frac{1}{2}\|[\mathbf{J}^{(0)}]^\top(\widetilde{\boldsymbol{\theta}}^{(0)} - \boldsymbol{\theta}^{(0)}) - \mathbf{v}\|_2^2 + \frac{m\lambda}{2}\|\widetilde{\boldsymbol{\theta}}^{(0)} - \boldsymbol{\theta}^{(0)}\|_2^2$$

$$\leq T/2,$$

where the first inequality holds trivially, the second inequality holds due to the monotonic decreasing property brought by gradient descent, the third inequality holds due to (C.6) in Lemma C.2. It is easy to verify that $\widetilde{L}$ is a $\lambda$-strongly convex and function and $C_1(tmL + m\lambda)$-smooth function, since

$$\nabla^2 \widetilde{L} \preceq (\|\mathbf{J}^{(0)}\|_2^2 + m\lambda)\mathbf{I} \preceq C_1(tmL + m\lambda),$$

where the first inequality holds due to the definition of $\widetilde{L}$, the second inequality holds due to (C.3) in Lemma C.2. Since we choose $\eta \leq C_2(tmL + m\lambda)^{-1}$ for some small enough $C_2 > 0$, then by standard result of gradient descent on ridge linear regression, $\widetilde{\boldsymbol{\theta}}^{(j)}$ converges to $\boldsymbol{\theta}^{(0)} + (\bar{\mathbf{Z}})^{-1}\bar{\mathbf{b}}/\sqrt{m}$ with the convergence rate

$$\big\|\widetilde{\boldsymbol{\theta}}^{(j)} - \boldsymbol{\theta}^{(0)} - (\bar{\mathbf{Z}})^{-1}\mathbf{b}/\sqrt{m}\big\|_2 \leq (1 - \eta m\lambda)^J\|\mathbf{f}^{(0)} - \mathbf{v}\|_2/\sqrt{m\lambda} \leq (1 - \eta m\lambda)^J\sqrt{t/(m\lambda)}.$$

$\qquad \square$

# E A VARIANT OF NEURALUCB

In this section, we present a variant of NeuralUCB called NeuralUCB$_0$. Compared with Algorithm 1, the main differences between NeuralUCB and NeuralUCB$_0$ are as follows: NeuralUCB uses gradient descent to train a deep neural network to learn the reward function $h(\mathbf{x})$ based on observed contexts and rewards. In contrast, NeuralUCB$_0$ uses matrix inversions to obtain parameters in closed forms. At each round, NeuralUCB uses the current DNN parameters ($\boldsymbol{\theta}_t$) to compute an upper confidence bound. In contrast, NeuralUCB$_0$ computes the UCB using the initial parameters ($\boldsymbol{\theta}_0$).

---

**Algorithm 3** NeuralUCB$_0$

---

1: **Input:** number of rounds $T$, regularization parameter $\lambda$, exploration parameter $\nu$, confidence parameter $\delta$, norm parameter $S$, network width $m$, network depth $L$
2: **Initialization:** Generate each entry of $\mathbf{W}_l$ independently from $N(0, 2/m)$ for $1 \leq l \leq L-1$, and each entry of $\mathbf{W}_L$ independently from $N(0, 1/m)$. Define $\boldsymbol{\phi}(\mathbf{x}) = \mathbf{g}(\mathbf{x}; \boldsymbol{\theta}_0)/\sqrt{m}$, where $\boldsymbol{\theta}_0 = [\text{vec}(\mathbf{W}_1)^\top, \ldots, \text{vec}(\mathbf{W}_L)^\top]^\top \in \mathbb{R}^p$
3: $\mathbf{Z}_0 = \lambda \mathbf{I}, \ \mathbf{b}_0 = \mathbf{0}$
4: **for** $t = 1, \ldots, T$ **do**
5:     Observe $\{\mathbf{x}_{t,a}\}_{a=1}^K$ and compute

$$(a_t, \widetilde{\boldsymbol{\theta}}_{t,a_t}) = \underset{a \in [K], \boldsymbol{\theta} \in \mathcal{C}_{t-1}}{\text{argmax}} \langle \boldsymbol{\phi}(\mathbf{x}_{t,a}), \boldsymbol{\theta} - \boldsymbol{\theta}_0 \rangle \tag{E.1}$$

6:     Play $a_t$ and receive reward $r_{t,a_t}$
7:     Compute

$$\mathbf{Z}_t = \mathbf{Z}_{t-1} + \boldsymbol{\phi}(\mathbf{x}_{t,a_t})\boldsymbol{\phi}(\mathbf{x}_{t,a_t})^\top \in \mathbb{R}^{p \times p}, \ \ \mathbf{b}_t = \mathbf{b}_{t-1} + r_{t,a_t}\boldsymbol{\phi}(\mathbf{x}_{t,a_t}) \in \mathbb{R}^p$$

8:     Compute $\boldsymbol{\theta}_t = \mathbf{Z}_t^{-1}\mathbf{b}_t + \boldsymbol{\theta}_0 \in \mathbb{R}^p$
9:     Construct $\mathcal{C}_t$ as

$$\mathcal{C}_t = \{\boldsymbol{\theta} : \|\boldsymbol{\theta}_t - \boldsymbol{\theta}\|_{\mathbf{Z}_t} \leq \gamma_t\}, \quad \text{where} \quad \gamma_t = \nu\sqrt{\log\frac{\det \mathbf{Z}_t}{\det \lambda\mathbf{I}} - 2\log\delta} + \sqrt{\lambda}S \tag{E.2}$$

10: **end for**

---

