# OpenReview forum: "NeuralUCB: Contextual Bandits with Neural Network-Based Exploration"
_ICLR.cc/2020/Conference — Reject_

### Official Review · AnonReviewer3 · 2019-10-14
**Official Blind Review #3**

**Rating:** 6

**Review:**

The authors proposed a neural network based UCB algorithm for bounded reward contextual bandit problems with theoretical guarantee thanks to the recent development of Neural Tangent Kernel (NTK).

Throughout the paper, the authors do not use much of the specific property of neural networks and NTK. They only use the gradient of neural networks as the feature and use the NTK as the kernel in the kernelized contextual bandits. This can be beneficial, for example, it can enriching the class of kernels. However, I feel the whole paper lacks novelty, and have some technical flaws.

Detailed Comments:
1. In Sec 3, the authors argued that kernelized contextual bandits suffers from the unknown RKHS problem and RKHS realizability problem. However, with universal kernel, RKHS is dense in L^2 space, thus can in principle approximate any function in L^2 space within any precision. So generally, this is not a problem. Moreover, bounded function does not necessarily contain the linear function, generalized linear function and bounded RKHS norm function. At least if we do not add some assumption on the input, linear function can be unbounded. On the other hand, the proposed methods also need p>TK to guarantee the realizability, and we can also design some kernel with feature map dimension larger than TK with some good property to guarantee realizability, so I think this claim is not fair.
2. In Assumption 5.2, the authors assume that the norm of contexts is smaller than 1. However, as far as I know, most of the existing work assumed the context have norm 1, and can be only relaxed to the norm upper lower bounded by two positive constant c1 and c2 (see [1]). Otherwise, there can be some issue on the positive definiteness of the NTK. Can the authors carefully check this? Meanwhile, I think it is not suitable to directly assume the NTK is positive definite. It is better to follow and refer the readers to the existing work.
3. It is better to introduce \theta^* before Sec 6, like for example the parameter that can perfectly predict the mean reward.
4. I am confusing on the proof of Lemma 6.1 in Page 12. When the authors calculate the norm of \theta^* - \theta_0, how to transform Q^\top A^{-2} Q to G^\top G? If we use the singular value decomposition, we only have that G^\top G=QA^2 Q^\top. If I understand correctly, here we should do an inverse, and we cannot simply get the desired results, as the minimum singular value of G can be small under current assumption. However, it is still possible to upper bound this distance to derive the remaining proof.
5. In the first line of Equation (B.3), there is a typo that omits the \phi(x)^\top.
6. How does the second inequality of (B.4) derives? I can understand that the authors may use the Cauchy-Schwartz inequality, but Frobenius norm cannot be directly upper bounded by spectral norm (though they are equivalent, but we need to add an additional constant like \sqrt{TK}). If the authors use the spectral norm, then the second term should be nuclear norm, not Frobenius norm. Probably I do not understand it correctly and it is not the core issue, but I think it is better to clarify it.
7. There is a typo in the fourth line of (B.4), it should be \lambda / \lambda_0,
8. The last derivation of Appendix B.3 have several typos omitting det(\lambda I).
9. The authors should better include the kernelized contextual bandits for a fair comparison, as LinUCB and Neural \epsilon-greedy both have theoretical issue that can be solved by kernel methods. I doubt that kernelized contextual bandits can solve these cases well.

Overall, I feel that most of the proof can be derived similarly from [2][3]. And Lemma C.1 is also from [4]. The authors only verify some conditions that when use NTK as the kernel in kernelized contextual bandits to adjust the main result from [2][3]. Thus, I think the technique used in this paper is not novel as well.

In my opinion, the communities are interested in solving contextual bandits with ``''gradient based'' neural network methods that use the neural network to predict the rewards given some contexts as input. But just as the Equation (6.1) shows, the prediction is not based on the neural network, but with a linear model taken \phi(x_i) as input. Also, throughout the paper, the authors never use the network output f. To this end, I feel this paper is over-claimed on ''neural''. On the other hand, I think it can be interesting to think about how kernel methods can benefit from NTK. Directly use the gradient as the feature map seems not an interesting and meaningful method, I think.

[1] Cao, Yuan, and Quanquan Gu. "A generalization theory of gradient descent for learning over-parameterized deep relu networks." arXiv preprint arXiv:1902.01384 (2019).
[2] Michal Valko, Nathan Korda, Rémi Munos, Ilias Flaounas, and Nello Cristianini. 2013. Finite-time analysis of kernelised contextual bandits. In Proceedings of the Twenty-Ninth Conference on Uncertainty in Artificial Intelligence (UAI'13), Ann Nicholson and Padhraic Smyth (Eds.). AUAI Press, Arlington, Virginia, United States, 654-663.
[3] Abbasi-Yadkori, Yasin, Dávid Pál, and Csaba Szepesvári. "Improved algorithms for linear stochastic bandits." Advances in Neural Information Processing Systems. 2011.
[4] Arora, Sanjeev, et al. "On exact computation with an infinitely wide neural net." arXiv preprint arXiv:1904.11955 (2019).

**Experience Assessment:**

I have read many papers in this area.

**Review Assessment: Checking Correctness Of Derivations And Theory:**

I carefully checked the derivations and theory.

**Review Assessment: Checking Correctness Of Experiments:**

I did not assess the experiments.

**Review Assessment: Thoroughness In Paper Reading:**

I read the paper thoroughly.

---

> ### Author Response · Authors · 2019-11-08
> **Response to Reviewer #3**
>
> Thank you for your feedback. We address your questions as follows. We have also revised our paper accordingly, and highlighted those places in blue.
>
>
> Q1: The statement of RKHS realizability problem is not fair.
> A1: Thanks for pointing this out.  We have removed such claims in the revision.
>
> Q2: The norm of contexts is smaller than 1.
> A2: Thanks for pointing out this minor issue. We have changed our assumption to $\|x_i\| = 1$.
>
> Q3. It is better to introduce $\theta^*$ before Sec 6, like for example the parameter that can perfectly predict the mean reward.
> A3: Thanks for the suggestion. $\theta^*$ is introduced only for the sake of analysis. It does not appear in our main results in Section 5. We have added a comment on $\theta^*$ right after Lemma 6.1 to explain its use.
>
> Q4: Confusing on the proof of Lemma 6.1
> A4: We have revised the proof of Lemma 6.1 and add more explanation in the proof.
>
> Q5: There exist some typos in the paper.
> A5: Thanks for pointing them out. We have corrected all the typos we found in the new version of this paper.
>
> Q6: How does the second inequality of (B.4) derives?
> A6: In our revision, the revised (B.4) is now (B.16). We have revised the inequality of (B.16), and add more explanation about its derivation.
>
> Q7: The authors should better include the kernelized contextual bandits
> A7: We will add the KernelUCB baseline in an upcoming revision.

---

> > ### Comment · AnonReviewer3 · 2019-11-08
> > **Thanks for your response. Below are some concerns after the main revision.**
> >
> > Thanks for your response. However, I still find there are some issues on the current version.
> >
> > 1. As $S\geq \sqrt{h H^{-1}h}$, it is still a issue of $S$. The authors can only argue that when $h$ projects to the spectral basis of $H$, there are no components on the eigenfunction corresponding to the small eigenvalue, which I think it’s still too restrictive. I don’t notice the mismatch of proof and the main text before, sorry for that. I also think this is a main drawback of the current theory on NTK, and it’s unfair to omit $S$ in the regret bound.
> > 2. I don’t think the core idea is far more different from the original version. Notice that m should be sufficient large and \lambda is larger than 1, so the regularization term dominates the update of the neural network and the parameter would not be far away from the initialization. Thus we can use the similar strategy, with an additional step bound the difference between the update version and no-update version, (I think just as the authors do). The relevant theorems are all proposed in [1] and [2], so I still find the technique is not novel.
> > 3. Can you address why NeuralUCB outperforms NeuralUCB_0 in the experiments in theory? Currently I don’t find it convincing, as if we over-parameterized the network, the neural tangent feature will not change more if the movement of the parameters is small.
> >
> > Minor comments:
> > 1. In theorem 5.5, the constant before the optimization error term should be C_2.
> > 2. The LHS of the conclusion in Lemma B.5 have some typos.
> > 3. Please polish the paper again to eliminate all of the possible typos like I mentioned before.
> >
> > I have raised my score to 3, as the authors exactly do some kinds of non-trivial thing that exactly (following the idea I mentioned before) to finish the whole proof. However, due to the reason I mentioned, I still tend to reject this paper. I don’t have enough time to go through the new proof, so there can be some misunderstanding on it. If the authors can convince me that the proof strategy is totally different from the strategy I mentioned that explicitly shows the newly proposed methods is significantly different to the previous NTK-KernelUCB, and show that the issue of S can be solved properly, I will consider raise the score to 6.
> >
> > [1] Cao, Yuan, and Quanquan Gu. "Generalization Bounds of Stochastic Gradient Descent for Wide and Deep Neural Networks." arXiv preprint arXiv:1905.13210 (2019).
> > [2] Allen-Zhu, Zeyuan, Yuanzhi Li, and Zhao Song. "A Convergence Theory for Deep Learning via Over-Parameterization." International Conference on Machine Learning. 2019.

---

> > > ### Author Response · Authors · 2019-11-11
> > > **To Reviewer 3’s new comment:**
> > >
> > > Q1: The RKHS norm of $h$ in Remark 5.8
> > > A1: We believe this is a misunderstanding about the definition of RKHS norm of $h$. Let $K$ be the NTK kernel function, then by the definition of $\mathbf{H}$, we have $\mathbf{H}_{i,j} = K(x^i, x^j)$. First, let us suppose the function $h$  can be represented by $h(x) = \sum_{i=1}^{TK} \alpha_i K(x, x^i)$, where $\{\alpha_i\}$’s are the coefficients. Then, according to Theorem 22 (page 118) in https://web.stanford.edu/class/cs229t/notes.pdf, the RKHS norm of $h$ is $\|h\|_{\mathcal{H}}=\sqrt{\sum_{i,j=1}^{TK} \alpha_i \alpha_j K(x^i, x^j)}= \sqrt{\mathbf{\alpha}^\top\mathbf{H}\mathbf{\alpha}}$, where $\mathbf{\alpha} = (\alpha_i,\dots,\alpha_{TK})^\top$. On the other hand, we can show that $\mathbf{\alpha} = \mathbf{H}^{-1}\mathbf{h}$ by substituting $x = x^i$ into $h(x) = \sum_{i=1}^{TK} \alpha_i K(x, x^i)$ and solving the system of linear equations. Then the RKHS norm of $h$ can be rewritten as $\|h\|_{\mathcal{H}} = \sqrt{\mathbf{\alpha}^\top\mathbf{H}\mathbf{\alpha}} = \sqrt{\mathbf{h}^\top\mathbf{H}^{-1}\mathbf{h}}$.  In the more general case where $h$ may not be represented by a weighted sum of kernel functions over a finite number of data points, we show in Appendix A.3 that the RKHS norm of $h$ can be lower bounded by $\sqrt{\mathbf{h}^\top \mathbf{H}^{-1}\mathbf{h}}$.
> > >
> > > Q2: I don’t think the core idea is far more different from the original version… lemmas can be found in [1] and [2].
> > > A2: Our proof structure is indeed similar to regret analyses of many UCB-based bandit algorithms, where a common key step is to find a proper confidence region/interval that vanishes fast enough.  That said, the proof is not trivial, and at various places requires different proof techniques.  For example,
> > > * We present a novel instantaneous regret bound in Lemm 6.3, which contains the function approximation error between the neural network function $f(x_{t,a}; \theta_{t-1})$ and its first-order approximation $f(x_{t,a}; \theta_{0}) + g(x_{t,a}; \theta_{t-1})^\top(\theta_{t-1} - \theta_0)$ as what we have shown in (B.10) in the proof of Lemma 6.3.  This bound is different from those in previous results such as Eq. (7) of Abbasi et al., (2011) and Lemma 1 of Valko et al., (2013).
> > > * We show in Lemma B.2 that applying gradient descent on the objective function $L$ at round $t$, the output parameter of TrainNN $\theta_t$ belongs to a neighborhood of the initial parameter $\theta_0$. While the lemma looks similar to previous work [1,2], its proof is quite different. In specific, our objective function $L$ has an extra term of $\ell_2$ regularization, thus its global minimum value is not zero, even the neural network is overparameterized. So the proof techniques in previous work [1,2] which highly rely on the zero global minimum are not directly applicable to our setting. We need to carry out a new analysis to show that GD still enjoys a linear rate of convergence for optimizing the regularized loss function $L$ to prove the statement of Lemma B.2.
> > >
> > > Q3: Can you address why NeuralUCB outperforms NeuralUCB_0 in the experiments in theory?
> > > A3: We believe the performance gap between NeuralUCB and NeuralUCB$_0$ is analogous to the gap between neural networks and the corresponding NTK in supervised learning, which has been observed and studied both in theory [3,4] and in practice [5].  However, even in the simpler case of supervised learning, a thorough understanding of this phenomenon remains an open problem, which is beyond the scope of this paper and will be an interesting topic for future work.
> > >
> > > Q4: Some typos.
> > > A4: Thanks for pointing them out. We have fixed these typos and polished other parts of the paper as well.
> > >
> > > [1] Yuan Cao, and Quanquan Gu. Generalization Bounds of Stochastic Gradient Descent for Wide and Deep Neural Networks. arXiv preprint arXiv:1905.13210 (2019).
> > > [2] Zeyuan Allen-Zhu, Yuanzhi Li, and Zhao Song. A convergence theory for deep learning via overparameterization. In ICML, 2019.
> > > [3] Zeyuan Allen-Zhu and Yuanzhi Li. What Can ResNet Learn Efficiently, Going Beyond Kernels? ArXiv e-prints, abs/1905.10337, May 2019.
> > > [4] Colin Wei, Jason D Lee, Qiang Liu, and Tengyu Ma. Regularization Matters: Generalization and Optimization of Neural Nets v.s. their Induced Kernel. arXiv preprint arXiv:1810.05369v3, 2019.
> > > [5] Sanjeev Arora, Simon S Du, Wei Hu, Zhiyuan Li, Ruslan Salakhutdinov, and Ruosong Wang. On exact computation with an infinitely wide neural net. arXiv preprint arXiv:1904.11955, 2019

---

### Official Review · AnonReviewer1 · 2019-10-23
**Official Blind Review #1**

**Rating:** 3

**Review:**

This paper proposes to use the Neural Tangent Kernel (NTK) with the Upper Confidence Bound for stochastic contextual bandits.
- The paper instantiates Kernel UCB (Valko, 2013) with the NTK and the novelty is limited from a theoretical point of view.
- There is no experimental comparison with Neural Linear or Kernel UCB using a fixed kernel, (for example, the RBF kernel) or to methods like Thompson sampling that work well in practice even with non-linearities.

Detailed review below:
- Section 2.2 is not relevant to the paper and it might be more useful to use this space to explain NTK better.
- Please explain NTK before instantiating the algorithm in Section 4.
- The "Efficient Implementation" section in Section 4 is standard and done in all the linear bandit papers. Please acknowledge this or say how it is different.
- The NTK description in Definition 5.1 needs to be clarified. At the moment, it is difficult to parse. Please give some intuition about it.
- For the regret analysis, could you explain how the analysis is different from that of a fixed kernel in Valko, 2013.
- What is the intuition for having a lower bound on "S", the norm parameter? Why is there no upper bound?
- The width of the neural network depends on T^4. How does this affect the effective dimension \tilde{d} in the worst case? Can it result in linear regret?
- For Lemma 6.2, 6.3, please say that these are directly borrowed from Valko, 2013 and Abbasi, 2011.
- From an experimental perspective, the width of the neural network is a constant wrt to T, K and L and clearly doesn't align with the theoretical bounds. Please justify why this is a valid thing to do?
- As mentioned earlier, there is no comparison with Kernel UCB with a fixed kernel, Neural Linear or Thompson sampling, methods that work well in practice.
- Finally, real-world experiments are necessary to show the benefit of using NTK in practice.

**Experience Assessment:**

I have published in this field for several years.

**Review Assessment: Checking Correctness Of Derivations And Theory:**

I assessed the sensibility of the derivations and theory.

**Review Assessment: Checking Correctness Of Experiments:**

I carefully checked the experiments.

**Review Assessment: Thoroughness In Paper Reading:**

I read the paper thoroughly.

---

> ### Author Response · Authors · 2019-11-08
> **Response to Reviewer #1**
>
> Thank you for your constructive feedback. We address your comments and questions as follows. We have also revised our paper accordingly, and highlighted those places in blue.
>
> Q1: "Section 2.2 is not relevant to the paper and it might be more useful to use this space to explain NTK better. "
> A1: We have added more explanation about NTK in the revision.  Neural tangent kernel (NTK) is originally defined in Jacob et al. (2018) by the gradient of the output of a randomly initialized neural network. In recent development of deep learning theory, a deep neural network can be characterized by its NTK in certain regime. Thus, we present the definition of NTK in Section 5.
>
> Q2: "The "Efficient Implementation" section in Section 4 is standard and done in all the linear bandit papers. Please acknowledge this or say how it is different. "
> A2: Thanks for your suggestion. This is indeed a standard technique. We have acknowledged this in the revision.
>
> Q3: "For the regret analysis, could you explain how the analysis is different from that of a fixed kernel in Valko, 2013. "
> A3: We have modified the NeuralUCB algorithm in the revision, so that it better reflects how DNNs are used to solve contextual bandits.  A key difference from the old version is that it now uses the most recent network parameter vector $\theta_t$, *not* the initial one $\theta_0$, to construct the upper confidence bound.  This makes the analysis substantially more challenging, and different from previous bandit analysis.  Specifically,
>   * Our previous algorithm (now called NeuralUCB$_0$ in Appendix E) can be regarded as KernelUCB with Neural Tangent kernel. However, our new algorithm NeuralUCB directly uses deep neural networks to predict the underlying reward function $h(x)$, which is less similar to KernelUCB due to its approximation error between the neural networks and their corresponding NTK kernel.
>   * Valko et al. (2013) analyzed the regret bound of a meta algorithm SupKernelUCB to handle the independence of rewards $r_{t, a_t}$, while we analyze the regret bound directly on NeuralUCB.
>
>
> Q4: "What is the intuition for having a lower bound on 'S', the norm parameter? Why is there no upper bound?"
> A4: S is tuning parameter in our algorithm, which in our proof needs to be chosen such that $\|\theta^*-\theta_0\|_2\leq S$. This is analogous to the condition $\|\theta^*\|_2\leq S$ in Theorem 2 of Abbasi-Yadkori et al. (2011). Our Lemma 6.1 suggests that $\|\theta^*-\theta_0\|_2 \leq \sqrt{h^T H^{-1}h}$. Therefore, in order to make $\|\theta^*-\theta_0\|_2 \leq S$ hold, it suffices to choose $S \geq \sqrt{h^\top H^{-1}h}$.
>
>
> Q5: " The width of the neural network depends on T^4. How does this affect the effective dimension $\tilde{d}$ in the worst case? Can it result in linear regret?"
> A5: The effective dimension $\tilde d$ is not related to the width of neural network due to definition 5.3, since it is only determined by the NTK matrix $H$. $\tilde d$ can be regarded as a measure of how quickly the eigenvalues of $H$ decay, and it only depends on $T$ logarithmically in some specific cases (Valko et al., 2013). Thus, the use of effective dimension will not result in linear regret.
>
>
> Q6: “For Lemma 6.2, 6.3, please say that these are directly borrowed from Valko, 2013 and Abbasi, 2011.”
> A6: In our revision, the new version of Lemmas 6.2 and 6.3 are not directly implied by the results in Valko et al. (2013) and Abbasi-Yadkori et al. (2011). We also cite Valko et al. (2013) and Abbasi-Yadkori et al. (2011) in the corresponding proofs of these two lemmas.
>
> Q7: "From an experimental perspective, the width of the neural network is a constant wrt to T, K and L and clearly doesn't align with the theoretical bounds. Please justify why this is a valid thing to do?"
> A7: While existing over-parameterized NN analyses give interesting insights on optimization and generalization, the bounds on the required network width $m$ are likely not tight. Therefore, in experiments we choose m to be relatively large (but not as large as theory requires).
>
> Q8: "There is no experimental comparison with Neural Linear or Kernel UCB using a fixed kernel"
> A8: We added the comparison between NeuralUCB and NeuralUCB$_0$ (which can be seen as Kernel UCB  with NTK kernel). The experiments suggest that NeuralUCB is better than NeuralUCB$_0$. We will add the KernelUCB with Gaussian kernel in an upcoming revision.

---

### Official Review · AnonReviewer2 · 2019-10-27
**Official Blind Review #2**

**Rating:** 6

**Review:**

This paper proposes Neural UCB for the neural-linear bandit setting. The main contribution of the paper is the theorem that the proposed method, Neural UCB, is guarantee to achieved a good regret bound, which for the first time extends bandits result to neural networks. Overall the paper is well written and easy to follow.

While the result of this paper seems to be interesting, the idea of the paper is simply combining a recent progress on the neural tangent kernel for overparametrized neural networks and a standard linear UCB algorithm.

The main concern I have is about the constant S in the regret bound. Note that this constant is an upper bound of \sqrt{h^T H h}, where h is in the dimension of TK and H is in the dimension of TK by TK. A naive bound for S could be sup-linear in T, which makes the bound vacuous. What would be a lower bound for \lambda_0 for eg. the setting in the experiments?

Other comment:
1. It should be explicitly stated somewhere in the paper that x_{t,k} are assumed to be deterministic. Thus \theta^* is deterministic. It is more important that \theta^* does NOT depends on a_t. Otherwise lemma 6.2 could be problematic.

=====================
Based on the new version of the paper and the discussions, I change the score to weak accept.

**Experience Assessment:**

I have read many papers in this area.

**Review Assessment: Checking Correctness Of Derivations And Theory:**

I assessed the sensibility of the derivations and theory.

**Review Assessment: Checking Correctness Of Experiments:**

I did not assess the experiments.

**Review Assessment: Thoroughness In Paper Reading:**

I read the paper at least twice and used my best judgement in assessing the paper.

---

> ### Author Response · Authors · 2019-11-08
> **Response to Reviewer #2**
>
> Thank you for your constructive comments. We address your questions as follows. We have also revised our paper accordingly, and highlighted those places in blue.
>
>
> Q1: "The main concern I have is about the constant S in the regret bound."
> A1: It is correct that in the worst case our bound will not be sublinear and may be dependent on $\lambda_0$. However, in remark 5.7 we have shown a specific case that when the reward function h belongs to the RKHS space induced by NTK with bounded norm $\|h\|$, then $\sqrt{h^\top H^{-1}h}$, the lower bound of S, is less than $\|h\|$, which is a constant independent of T and K.
>
> Q2: "It should be explicitly stated somewhere in the paper that x_{t,k} are assumed to be deterministic. Thus \theta^ is deterministic. It is more important that \theta^ does NOT depends on a_t. Otherwise lemma 6.2 could be problematic."
> A2: We believe this is a misunderstanding. Our analysis does not require $\{x_{t,a}\}$ to be deterministic, which can be demonstrated as follows. In our revision, the revised Lemma 6.2 now became Lemma 6.1. Due to the proof of Lemma 6.1, it can be seen that $\theta^* = \theta_0 + PA^{-1}Q^\top h/\sqrt{m}$, where $PAQ^\top$ is the SVD of $G$, $G = [g(x^1; \theta_0)\dots g(x^{TK}; \theta_0)]$. By the definition of $\theta^*$, it can be seen that:
> * $\theta^*$ is not deterministic since $\theta^*$ depends on $\theta_0$.
> * $\theta^*$ does not depend on $a_t$ because $G$ does not depend on $a_t$.
> Therefore, we do not require $\{x_{t,a}\}$ to be deterministic, as long as they are independent of $\theta_0$.

---

> > ### Comment · AnonReviewer2 · 2019-11-15
> > **After Rebuttal**
> >
> > Thanks for the response.
> >
> > 1) I am not sure how restrictive this assumption would be that h is a function in the unit ball (let's say constant RKHS norm) in the RKHS space of the NTK. How representative this space is, given NTK. I am not familiar with the literature of NTK but I think it needs further justification that this setting still has enough modelling capability.
> >
> > 2) The reason of saying $x_{t,a}$ being deterministic: to apply the result of Abbasi-Yadkori et al., 2011, the setting in this paper has to be the same with that in the referred paper. Note that \theta^* depends on $x_{t,a}$ which is not the case in Abbasi-Yadkori et al., 2011. Here it needs more careful checking if the result in  Abbasi-Yadkori et al., 2011 is applicable or not.

---

> > > ### Author Response · Authors · 2019-11-15
> > > **To Reviewer 2’s new comments**
> > >
> > > 1): “I am not familiar with the literature of NTK but I think it needs further justification that this setting still has enough modelling capability.”
> > >
> > > The function class induced by NTK has a strong representative power. For instance, recent work [1,2,3] have shown that a variety of NTKs are able to achieve comparable results as the corresponding neural networks on different tasks on CIFAR10, UCI database, VOC07, and achieve better results than Gaussian kernel methods. We will add this justification in the final version of our paper.
> > >
> > >
> > > 2): “x_{t,a} being deterministic”
> > >
> > > We have carefully checked the entire proof in Abbasi-Yadkori et al., 2011, and confirmed that their proof does not require $\theta^*$ to be independent of $x_{t,a}$. They only require $\|\theta^*\|_2 \leq S$. Therefore, we can indeed use the corresponding result in Abbasi-Yadkori et al., 2011 in our proof, and our analysis is correct even if $x_{t,a}$’s are not deterministic.
> > >
> > > [1] Arora, S., Du, S. S., Hu, W., Li, Z., Salakhutdinov, R., & Wang, R. (2019). On exact computation with an infinitely wide neural net. arXiv preprint arXiv:1904.11955.
> > > [2] Arora, S., Du, S. S., Li, Z., Salakhutdinov, R., Wang, R., & Yu, D. (2019). Harnessing the Power of Infinitely Wide Deep Nets on Small-data Tasks. arXiv preprint arXiv:1910.01663.
> > > [3] Li, Z., Wang, R., Yu, D., Du, S. S., Hu, W., Salakhutdinov, R., & Arora, S. (2019). Enhanced Convolutional Neural Tangent Kernels. arXiv preprint arXiv:1911.00809.

---

> > > > ### Comment · AnonReviewer2 · 2019-11-15
> > > > **Thanks for the quick response**
> > > >
> > > > 1) Is NTK a universal kernel? What is its approximation power? But given the evidence you provided in your response, I am not very concerned about this question now.
> > > >
> > > > 2) I do think the analysis in Abbasi-Yadkori et al., 2011 requires \theta^* to be a constant not depending on x_{t,a}. I will double check with the paper of Abbasi-Yadkori et al., 2011. I understand the deadline for rebuttal is approaching. I will adapt my rating accordingly after I check the correctness of the response.

---

> > > > > ### Author Response · Authors · 2019-11-15
> > > > > **Thank you**
> > > > >
> > > > > 1) Is NTK a universal kernel?
> > > > >
> > > > > This very recent paper (https://arxiv.org/abs/1910.06956 ) establishes rates of universal approximation for the shallow neural tangent kernel (NTK).

---

### Public Comment · ~Sean_Kanne1 · 2019-10-05
**Not Neural Network, it is just linear contextual bandit with random features**

It seems the algorithm does not utilize any feature of deep neural network. The authors directly use the neural tangent kernel in the algorithm. This makes the algorithm a direct application of linear UCB with features functions given by the random features produced by the neural tangent features. Thus, it would be better to change the name to ``"contextual bandit with random features".

---

### Public Comment · ~Anna_Dudley1 · 2019-10-06
**exactly random feature (neural tangent feature); proof almost identical to Valko et al. (2013)**

(I am adding to Sean Kanne's comment. I was attracted to the title but got disappointed by the paper.)

Despite the very attractive title, this paper is simply restating a special case of kernelized contextual bandit, which is originally developed by Valko et al. (2013). In particular, this paper specializes the kernel of Valko et al. (2013) to be the neural tangent kernel, which is induced by the random feature corresponding to the initial weights. Given this fact, the whole paper can be summarized into a half-page corollary of Valko et al. (2013).

Interestingly, the whole algorithm in this paper does not use SGD to train a neural network at all. Instead, it just uses the random feature corresponding to the initial weights to perform linear regression. I am not sure if it is proper to call it "neural", which is very misleading.

Valko et al. (2013), Finite-Time Analysis of Kernelised Contextual Bandits.

---

### Author Response · Authors · 2019-11-08
**[To all reviewers] Major change in revised version**

Thank you very much for your constructive comments. We have uploaded a revised version of our paper.

In the revision, we modified the NeuralUCB algorithm, while the original algorithm in the initial submission will be referred to as NeuralUCB$_0$.  The new algorithm better reflects how DNNs are used to solve contextual bandits in practice.

The main differences between NeuralUCB and NeuralUCB$_0$ are:
*NeuralUCB uses gradient descent to train a deep neural network to learn the reward function $h(x)$ based on observed contexts and rewards.  In contrast, NeuralUCB$_0$ uses matrix inversions to obtain parameters in closed forms.
*At each round, NeuralUCB uses the current DNN parameters ($\theta_t$) to compute an upper confidence bound.  In contrast, NeuralUCB$_0$ computes the UCB using the initial parameters ($\theta_0$).
*We compared NeuralUCB with NeuralUCB$_0$ empirically in Section 7. NeuralUCB outperforms NeuralUCB$_0$ in all experiment settings, which suggests that the adaptive feature mapping brought by NeuralUCB is better than fixed feature mapping used by NeuralUCB$_0$.

---

### Decision · Program_Chairs · 2019-12-19

**Decision:**

Reject

**Comment:**

As the reviewers have pointed out and the authors have confirmed, the original version of this paper was not a significant leap beyond combining recent understanding of Neural Tangent Kernels and previous techniques for kernelized bandits. In a revision, the authors updated their draft to allow the point at which gradients are centered around, theta_0, to now equal theta_t. This seems like a more reasonable algorithm and it is satisfying that the authors were able to maintain their regret bound for this dynamic setting. However, the revision is substantial and it seems unreasonable to expect reviewers to read the revised results in detail--the reviewers also felt it may be unfair to other ICLR submissions. All reviewers believe the paper has introduced valuable contributions to the area but should go under a full review process at a future venue. A reviewer would also like to see a comparison to Kernel UCB run on the true NTK (or a good approximation thereof).